# A conserved mechanism for regulating replisome disassembly in eukaryotes

Michael Jenkyn-Bedford[1,4], Morgan L. Jones[1,4], Yasemin Baris[1], Karim P. M. Labib[2], Giuseppe Cannone[1], Joseph T. P. Yeeles[1✉] & Tom D. Deegan[2,3✉]

Replisome disassembly is the final step of eukaryotic DNA replication and is triggered by ubiquitylation of the CDC45–MCM–GINS (CMG) replicative helicase[1–3]. Despite being driven by evolutionarily diverse E3 ubiquitin ligases in different eukaryotes (SCF[Dia2] in budding yeast[1], CUL2[LRR1] in metazoa[4–7]), replisome disassembly is governed by a common regulatory principle, in which ubiquitylation of CMG is suppressed before replication termination, to prevent replication fork collapse. Recent evidence suggests that this suppression is mediated by replication fork DNA[8–10]. However, it is unknown how SCF[Dia2] and CUL2[LRR1] discriminate terminated from elongating replisomes, to selectively ubiquitylate CMG only after termination. Here we used cryo-electron microscopy to solve high-resolution structures of budding yeast and human replisome–E3 ligase assemblies. Our structures show that the leucine-rich repeat domains of Dia2 and LRR1 are structurally distinct, but bind to a common site on CMG, including the MCM3 and MCM5 zinc-finger domains. The LRR–MCM interaction is essential for replisome disassembly and, crucially, is occluded by the excluded DNA strand at replication forks, establishing the structural basis for the suppression of CMG ubiquitylation before termination. Our results elucidate a conserved mechanism for the regulation of replisome disassembly in eukaryotes, and reveal a previously unanticipated role for DNA in preserving replisome integrity.

The eukaryotic replisome is assembled around the CMG helicase at replication origins during replication initiation. Once assembled, CMG remains stably associated with replication forks until two forks emanating from adjacent origins converge, or a single fork encounters the end of a linear chromosome or a template discontinuity, at which point replication terminates (Fig. 1a). Upon termination, the replisome is disassembled in two steps: first, CMG is ubiquitylated on its Mcm7 subunit by a cullin-RING E3 ubiquitin ligase (SCF[Dia2] in budding yeast, CUL2[LRR1] in metazoa); second, ubiquitylated Mcm7 is unfolded by the Cdc48 ATPase (also known as p97 in higher eukaryotes), leading to disassembly of the replisome[1–3,8–10]. As there is no known mechanism for origin-independent CMG assembly in S phase, premature disassembly of CMG must be avoided, to prevent replication fork collapse and genome instability[11]. CMG translocates on the leading-strand template while excluding the lagging-strand template from its central channel[12]. It has been suggested that this 'excluded' DNA strand, which is lost upon termination (Fig. 1a), inhibits ubiquitylation of CMG at replication forks[8–10]. However, because there are currently no structures of terminated replisomes in complex with SCF[Dia2] or CUL2[LRR1], how ubiquitylation of CMG is regulated to restrict replisome disassembly to termination remains a key unanswered question.

## Terminated yeast replisome structures

To determine the molecular basis for the regulation of CMG ubiquitylation, we aimed to solve the structure of a terminated replisome, by adapting our system for reconstituting budding yeast replisomes for structural analysis[13]. After convergence of two replication forks, CMG translocates onto nascent double-stranded DNA (dsDNA) produced by the converging replisome[3,9,14] (Fig. 1a). To trap a replisome bound around dsDNA, we used a DNA substrate that lacked a 5′ flap and contained a short stretch methylphosphonate modifications embedded in dsDNA, which slow translocation of CMG[15] (Extended Data Fig. 1a). This DNA substrate was incubated with CMG, the replisome factors Tof1–Csm3, Mrc1 and Ctf4, SCF[Dia2] (Hrt1–Cdc53–Skp1–Dia2), an E2–ubiquitin conjugate (Cdc34–Ub)[16], and the leading-strand DNA polymerase Pol-ε, in the presence of ATP (Extended Data Fig. 1a). After glycerol gradient sedimentation, complexes containing all replisome and SCF[Dia2] subunits were isolated (Extended Data Fig. 1b). Cdc34–Ub did not associate with the complex, perhaps reflecting the absence of neddylation on the Cdc53 cullin subunit of SCF[Dia2] (refs. [17,18]).

After gradient fixation, samples were prepared for cryo-electron microscopy (cryo-EM), yielding three-dimensional (3D) reconstructions at average resolutions of 3.2–4.0 Å (Fourier shell correlation (FSC) = 0.143 criterion; Extended Data Fig. 1c–h, Extended Data Table 1). DNA binding was heterogenous across the dataset, with the majority of particles still engaging single-stranded DNA (Extended Data Fig. 2). Nonetheless, we identified a subset of particles, which was subsequently subclassified into two conformations (conformations I and II), that had unambiguously translocated onto dsDNA, representative of bona fide termination intermediates produced after fork convergence

[1]MRC Laboratory of Molecular Biology, Cambridge, UK. [2]MRC Protein Phosphorylation and Ubiquitylation Unit, Sir James Black Centre, School of Life Sciences, University of Dundee, Dundee, UK. [3]Present address: MRC Human Genetics Unit, Institute of Genetics and Cancer, University of Edinburgh, Western General Hospital, Edinburgh, UK. [4]These authors contributed equally: Michael Jenkyn-Bedford, Morgan L. Jones. ✉e-mail: jyeeles@mrc-lmb.cam.ac.uk; tdeegan@ed.ac.uk

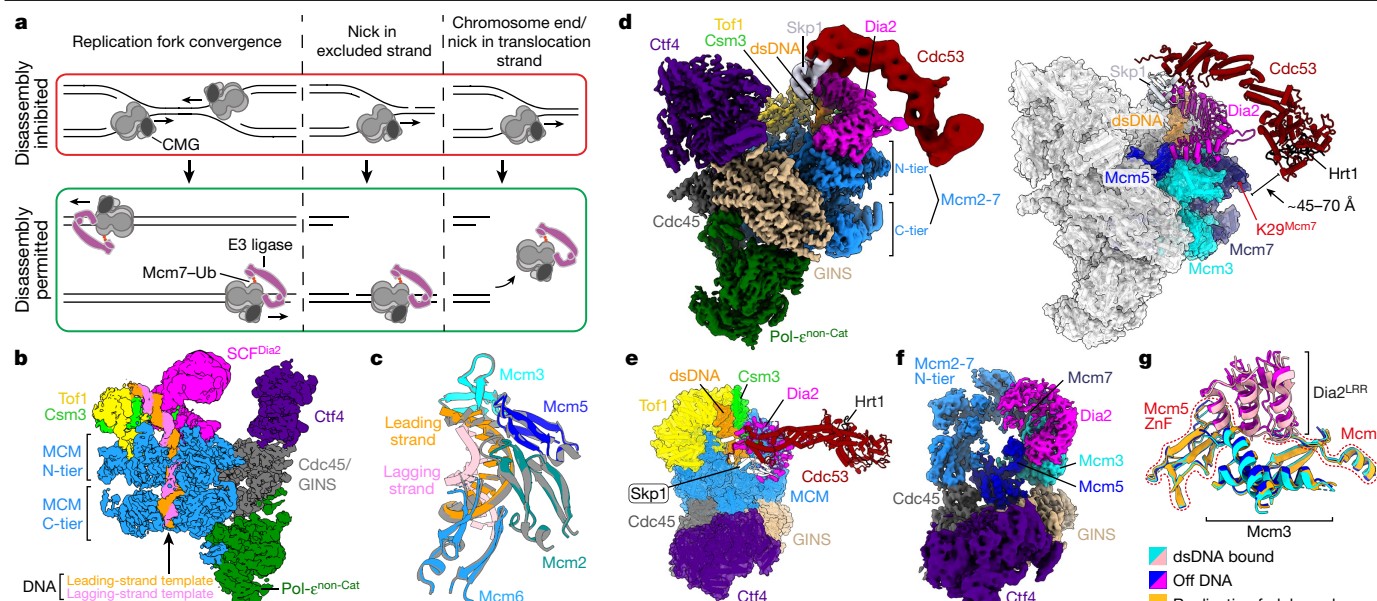

**Fig. 1 | Cryo-EM structures of terminated replisomes from *Saccharomyces cerevisiae* bound by SCF[Dia2]. a**, Schematic of the regulation of replisome disassembly. For clarity, replisomes are depicted as CMG. CMG ubiquitylation and replisome disassembly are inhibited at replication forks by an as yet unknown mechanism, dependent on the excluded DNA strand (in red box). This inhibition is relieved following translocation onto dsDNA (in green box, left and middle) or off DNA (in green box, right). **b**, Slice-through view of cryo-EM density for complexes assembled on dsDNA. The density shown is a composite of focused maps (refer to Extended Data Fig. 2). **c**, DNA engagement within the MCM C-tier motor domains by complexes assembled on dsDNA (coloured) or on a replication fork (grey; PDB: 6SKL[13]). **d**, Cryo-EM density as in **b** (left) and corresponding atomic model (right) for complexes assembled on dsDNA. For the atomic model, only SCF[Dia2], DNA and MCM subunits that interact with SCF[Dia2] are coloured. **e**, Alternative view of the atomic model in **d**. **f**, Cryo-EM density for complexes assembled in the absence of DNA, derived from multibody refinement. **g**, Comparison of the MCM–Dia2[LRR] interface from complexes assembled on dsDNA (**b**–**e**), off DNA (**f**) or on a replication fork (PDB: 6SKL[13]). For the regions of MCM at this interface, the root mean square deviation (r.m.s.d.) of the replication fork-bound complex compared with the dsDNA-bound or off-DNA complexes is 1.39 Å and 0.93 Å, respectively.

(Fig. 1b, Extended Data Figs. 2, 3a–e). While the configuration of the MCM C-tier differed between conformations I and II (Extended Data Fig. 3), in both cases the incoming dsDNA was bent by approximately 46° between the MCM N-tier and C-tier, necessitating distortion of the DNA duplex within the N-tier (Extended Data Fig. 3f). For conformation I, the nucleotide occupancy and interactions with the phosphate backbone of the leading-strand template are similar to replication fork-bound CMG[13] (Fig. 1c, Extended Data Fig. 3b, g), suggesting a shared mechanism for translocation of CMG over single-stranded DNA and dsDNA[15].

Having identified particles that had translocated onto dsDNA, we were able to build an atomic model of a terminated replisome (Fig. 1d). The overall architecture of CMG, Ctf4, Tof1–Csm3 and the non-catalytic module of Pol-ε (Pol-ε[non-Cat]) was almost indistinguishable from previous structures[13,19–21] (for details of the structure of Pol-ε, see Extended Data Fig. 4). We observed an additional, large region of density at the N-tier face of CMG beside Mcm3 and Mcm7, which closely approaches Csm3 and the dsDNA ahead of CMG, before extending away from the core of the complex, forming an elongated arm characteristic of the cullin subunit (Cdc53) of SCF[Dia2] (Fig. 1d, e). The resolution of the cullin arm is relatively poor (precluding model building for Cdc53–Hrt1), due to a large degree of flexibility in this region, as highlighted by comparison of 3D classes (Extended Data Fig. 5a). We predict that this flexibility is important for conjugating the long K48-linked polyubiquitin chains required for Cdc48-dependent replisome disassembly[8]. Regardless, the orientation of SCF[Dia2] can be unambiguously defined, placing the Cdc53 C terminus and Hrt1 ~45–70 Å from the primary ubiquitylation site on Mcm7 (Lys29)[8,22] (Extended Data Fig. 5a, b), consistent with previous structures of un-neddylated cullin-RING E3 ligases[23].

Density corresponding to the E3 ligase substrate-recognition module (Skp1–Dia2) is adjacent to the N-tier face of CMG (Fig. 1d, e). The N-terminal tetratricopeptide repeat domain of Dia2, which binds Ctf4 and Mrc1 (refs. [8,24,25]), was not visible in our structure. However, clear

secondary structure and side chain density enabled us to build a de novo atomic model for the remainder of Dia2, encompassing the F-box (residues 211–247), 15 tandem leucine-rich repeats (LRRs) (248–716) and a C-terminal tail (717–732), which folds back onto the concave surface of the horseshoe-shaped LRRs (Extended Data Fig. 5c–k). The C-terminal end of the LRR domain forms an extensive interface with the N-tier of the Mcm3, Mcm5 and Mcm7 subunits of CMG (Fig. 1d, Extended Data Fig. 5l–n; see text below for a detailed description), demonstrating that Dia2 binds directly to CMG bound around dsDNA, equivalent to the situation after convergence of two replication forks.

When DNA replication terminates at the end of linear chromosomes, CMG is thought to dissociate from DNA, at which point the loss of the excluded strand triggers CMG ubiquitylation[8,9] (Fig. 1a). To establish how SCF[Dia2] engages the replisome following termination at chromosome ends, we repeated cryo-EM sample preparation as described above, except in the absence of DNA. This yielded a 3D reconstruction of an 'off DNA' replisome at 3.9 Å resolution (Fig. 1f, Extended Data Fig. 6). Notably, binding of the Dia2 LRRs across Mcm3, Mcm5 and Mcm7 is indistinguishable from complexes bound around dsDNA (Fig. 1g, Extended Data Fig. 3e). Furthermore, comparison of our dsDNA-bound and off-DNA complexes with a previous structure of a replication fork-associated replisome[13] revealed no conformational changes in the region of the MCM N-tier to which Dia2 binds (Fig. 1g, Extended Data Fig. 3e). Therefore, we conclude that termination does not induce conformational changes in CMG that are important for the regulation of CMG ubiquitylation by SCF[Dia2] (ref. [26]), either following fork convergence or when CMG dissociates from DNA.

## Dia2[LRR]–MCM interface

The extensive interface between Dia2[LRR] and MCM is predominantly formed by the Mcm3 N-tier (helices α1 and α5 and the zinc-finger (ZnF)

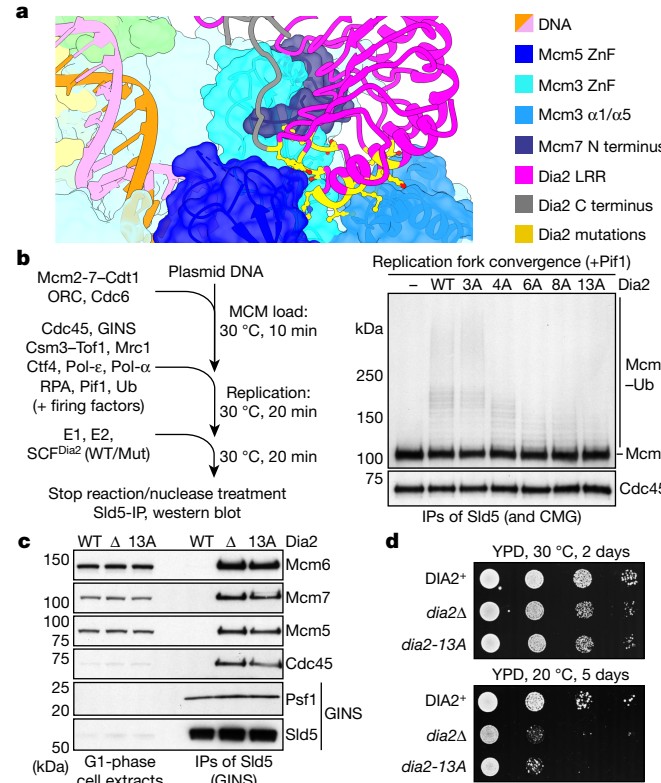

**a**

Legend:
- DNA (orange/pink)
- Mcm5 ZnF (blue)
- Mcm3 ZnF (cyan)
- Mcm3 α1/α5 (light blue)
- Mcm7 N terminus (dark blue)
- Dia2 LRR (magenta)
- Dia2 C terminus (grey)
- Dia2 mutations (yellow)

**b**

Mcm2-7–Cdt1 ORC, Cdc6 → Plasmid DNA
MCM load: 30 °C, 10 min
↓
Cdc45, GINS Csm3–Tof1, Mrc1 Ctf4, Pol-ε, Pol-α RPA, Pif1, Ub (+ firing factors)
Replication: 30 °C, 20 min
↓
E1, E2, SCF^Dia2 (WT/Mut)
30 °C, 20 min
↓
Stop reaction/nuclease treatment Sld5-IP, western blot

Replication fork convergence (+Pif1)

| | – | WT | 3A | 4A | 6A | 8A | 13A | Dia2 |
kDa
250
150
100
75

Mcm7 –Ub
Mcm7
Cdc45

IPs of Sld5 (and CMG)

**c**

| | WT | Δ | 13A | | WT | Δ | 13A | Dia2 |
150 — Mcm6
100 — Mcm7
100 / 75 — Mcm5
75 — Cdc45
25 / 20 — Psf1 (GINS)
50 — Sld5

(kDa) G1-phase cell extracts | IPs of Sld5 (GINS)

**d**

YPD, 30 °C, 2 days
DIA2+
dia2Δ
dia2-13A

YPD, 20 °C, 5 days
DIA2+
dia2Δ
dia2-13A

**Fig. 2 | The MCM–Dia2^LRR interface is required for replisome disassembly.**
**a**, Overview of the MCM–Dia2^LRR interface. Leading-strand and lagging-strand template DNA is coloured orange and pink, respectively. Residues altered in Dia2^LRR mutants are in yellow. **b**, Reaction scheme to monitor CMG–Mcm7 ubiquitylation after Pif1-stimulated replication fork convergence in vitro[14] (left). Immunoblot of reactions conducted as indicated is also shown (right). The experiment was repeated three times. IP, immunoprecipitation; Mut, mutant; Ub, ubiquitin; WT, wild type. **c**, SDS–PAGE and immunoblotting of TAP–Sld5 immunoprecipitations from G1-arrested yeast cells with the indicated Dia2 alleles. The experiment was repeated twice. Also see Extended Data Fig. 7i. TAP, tandem affinity purification. **d**, Spot-dilution assay (tenfold serial dilutions) with the indicated yeast strains. The experiment was repeated three times. For gel source data, see Supplementary Fig. 1. YPD, yeast extract peptone dextrose.

domain), which forms a cradle for the C terminus of Dia2^LRR (Fig. 2a, Extended Data Fig. 5l–n). In addition, the N terminus of Mcm7 wraps around the ZnF domain of Mcm3 and becomes sandwiched between

Mcm3 and Dia2, while the ZnF domain of Mcm5 interacts with the C-terminal end of Dia2^LRR, at the periphery of the Dia2^LRR–MCM interface. The details of the residues involved are illustrated in Extended Data Fig. 5n.

To examine the significance of the Dia2^LRR–MCM interaction for CMG ubiquitylation and replisome disassembly, we generated a series of point mutants targeting the Dia2^LRR–MCM interface, in both Dia2 (Fig. 2a) and MCM. The majority of MCM mutants exhibited defects in the formation of the Mcm2-7–Cdt1 complex or in MCM loading (data not shown), probably because the Dia2 LRR binding site is positioned at the inter-hexamer interface in the MCM double hexamer[27]. While this precluded analyses of Mcm7 ubiquitylation after convergence of two replication forks in vitro, we were able to purify a CMG complex containing mutations in Mcm3 and Mcm5, which, while being proficient for DNA replication, was defective for ubiquitylation of Mcm7 (Extended Data Fig. 7a–d). Dia2^LRR mutants formed stable tetrameric SCF^Dia2 complexes and supported ubiquitylation of Ctf4 (Extended Data Fig. 7e–g). Importantly, with the exception of Dia2-3A, the Dia2^LRR mutants were defective for ubiquitylation of Mcm7, both after replication fork convergence (Fig. 2b) and off DNA (Extended Data Fig. 7h), with Dia2-13A showing the most penetrant defect. Haploid yeast cells with the *dia2-13A* allele accumulated CMG in the G1 phase of the cell cycle (Fig. 2c, Extended Data Fig. 7i), reflecting a failure to disassemble CMG during replication termination in the S phase of the previous cell cycle[1]. Furthermore, these cells exhibited a profound growth defect at 20 °C, indistinguishable from cells lacking Dia2 (ref. [24]) (Fig. 2d). Together, these data demonstrate that the Dia2^LRR–MCM interface that we describe is essential for CMG ubiquitylation and replisome disassembly, both after fork convergence and when CMG dissociates from DNA.

## Human replisome–CUL2^LRR1 structure

Ubiquitylation of CMG in metazoa is driven by CUL2^LRR1 (LRR1–CUL2– ELOB–ELOC–RBX1)[4–7]. Although LRR1 displays no apparent sequence homology to Dia2, metazoan CUL2^LRR1 ubiquitylates CMG on its MCM7 subunit[4–6] and is suppressed by the excluded DNA strand[9,10], suggesting there might be common features of replisome association that are important for the regulation of both SCF^Dia2 and CUL2^LRR1. To investigate this, we used our approach for human replisome assembly[28] and a DNA substrate lacking a 5′ flap, to determine a high-resolution structure of CUL2^LRR1 in the human replisome (Fig. 3a, b, Extended Data Figs. 8, 9).

The overall architecture of human CMG, AND-1, TIMELESS–TIPIN and Pol-ε are indistinguishable from our previous structure lacking CUL2^LRR1 (ref. [28]) (Fig. 3a, b, Extended Data Fig. 10a). LRR1 is positioned across the MCM N-tier, in close proximity to the parental DNA duplex. In addition, an elongated arm of lower-resolution density, into which the

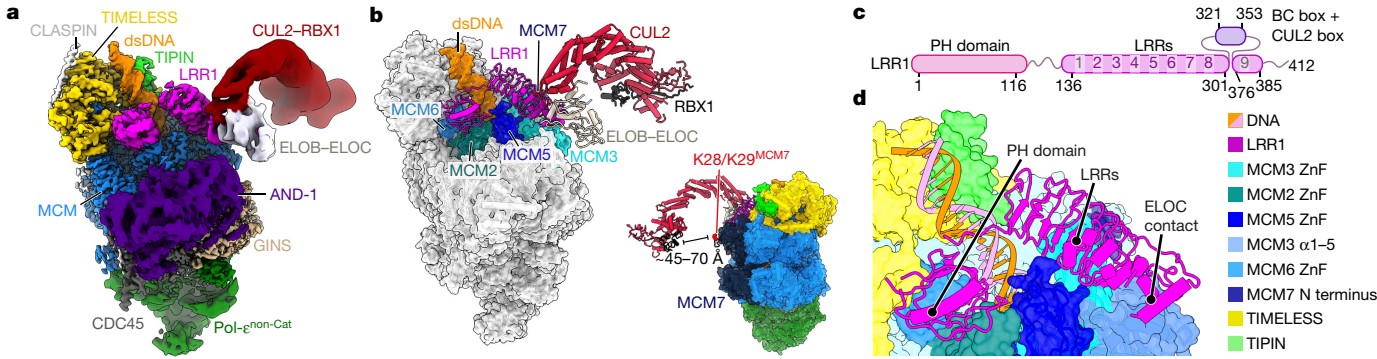

**a**
CLASPIN, TIMELESS, dsDNA, TIPIN, LRR1, CUL2–RBX1, ELOB–ELOC, MCM, AND-1, GINS, CDC45, Pol-ε^non-Cat

**b**
dsDNA, MCM7, LRR1, CUL2, MCM6, MCM2, MCM5, MCM3, RBX1, ELOB–ELOC, K28/K29^MCM7, 45–70 Å, MCM7

**c**
LRR1
PH domain (1–116), LRRs (1 2 3 4 5 6 7 8 9), 136, 301, 376, 385, 321 353 BC box + CUL2 box, 412

**d**
PH domain, LRRs, ELOC contact

Legend:
- DNA (orange/pink)
- LRR1 (magenta)
- MCM3 ZnF (cyan)
- MCM2 ZnF (teal)
- MCM5 ZnF (blue)
- MCM3 α1–5 (light blue)
- MCM6 ZnF (light blue)
- MCM7 N terminus (dark blue)
- TIMELESS (yellow)
- TIPIN (green)

**Fig. 3 | Cryo-EM structures of human replisomes bound by CUL2^LRR1. a**, Cryo-EM density of the human replisome bound by CUL2^LRR1. The density shown is a composite of focused maps (refer to Extended Data Fig. 8). **b**, Atomic models for the human replisome bound by CUL2^LRR1 displayed using transparent surface rendering, except for CUL2^LRR1. Only CUL2^LRR1, DNA and the CUL2^LRR1-interacting

regions of MCM are coloured (left). The model indicating the distance between RBX1 and K28/K29^MCM7 is coloured according to subunit (right). **c**, LRR1 domain architecture diagram. The primary sequence and LRRs 1–9 are numbered. PH, pleckstrin homology. **d**, Overview of the interface between LRR1 and the replisome. The model is displayed using surface rendering, except for LRR1 and DNA.

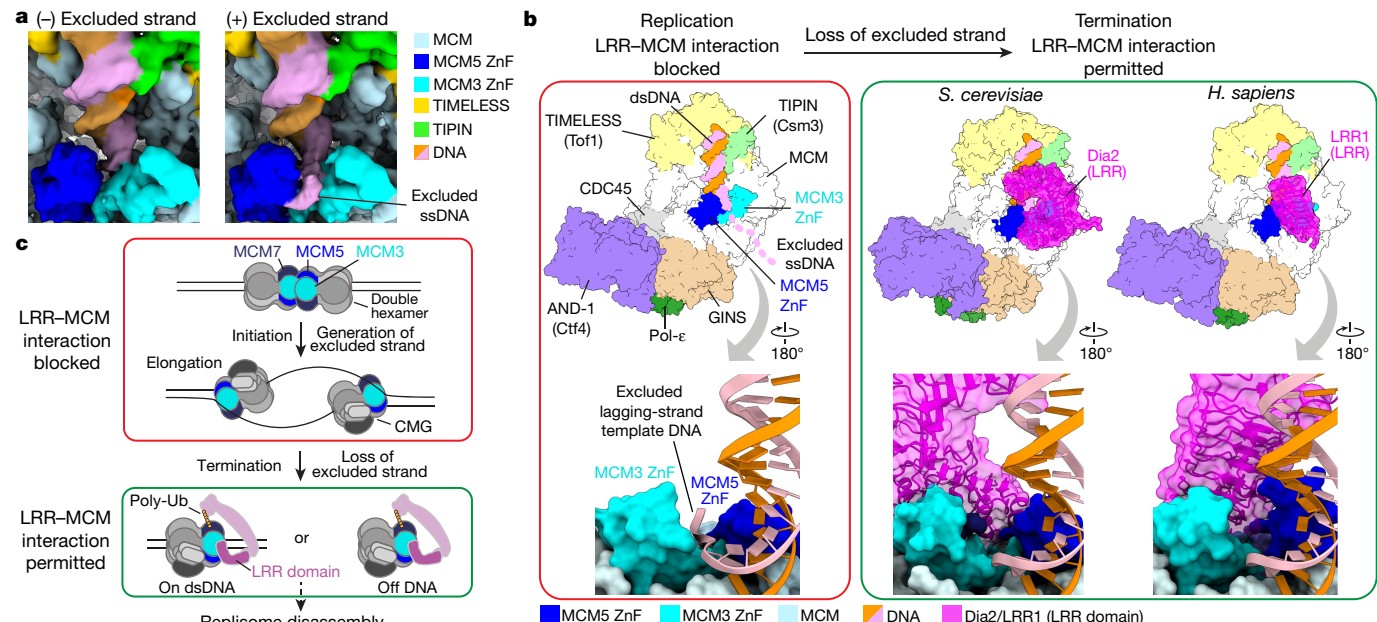

**Fig. 4 | A conserved mechanism for regulating replisome disassembly in eukaryotes. a**, Comparison of cryo-EM density maps for human replisome complexes (CMG, TIMELESS, TIPIN, CLASPIN, AND-1 and Pol-ε) bound to DNA substrates either lacking (left) or featuring (right; EMDB: EMD-13375 (ref. [28])) a 15-nucleotide 5′ flap, representing the excluded DNA strand. Density is coloured according to chain occupancy using a radius of 5 Å, with the excluded strand coloured manually in UCSF Chimera. ssDNA, single-stranded DNA. **b**, Alternative views of the ZnF domains of MCM3 and MCM5 during replication elongation (red box, excluded strand present[28]) and termination (green box, excluded strand absent). In the upper panel of the red box, the dashed line shows a putative path for the excluded ssDNA beyond the density observed in **a**, right. In the lower panel of the red box, four sugar-phosphate backbone linkages were built into the excluded strand density (see **a**, right; EMDB: EMD-13375 (ref. [28])). *H. sapiens*, *Homo sapiens*. **c**, Model for the regulation of CMG ubiquitylation. LRR-interacting regions of MCM are occluded in the MCM double hexamer (see Extended Data Fig. 11a) and by the excluded DNA strand at replication forks (see **a**, **b**) (red box). Loss of the excluded strand upon termination allows LRR–MCM engagement, CMG ubiquitylation and replisome disassembly (green box).

crystal structure of ELOB–ELOC–CUL2–RBX1 could be unambiguously docked[29], projects from the MCM N-tier in an analogous manner to yeast Cdc53–Hrt1 (Fig. 3b). Although metazoan CUL2 and yeast Cdc53 are tethered to their respective substrate adaptors (ELOB–ELOC–LRR1 for CUL2, Skp1–Dia2 for Cdc53) via very different interactions, the cullin C terminus and RING-box protein are similarly located in both cases, ~45–70 Å from the primary ubiquitylation sites in Mcm7 (refs. [8,9,22]) (Figs. 1d, 3b). Furthermore, like Cdc53, CUL2 displays considerable conformational variability, which is probably important for the conjugation of long polyubiquitin chains onto MCM7 (refs. [8,30]) (Extended Data Figs. 5a, 10b).

The majority of LRR1 was well resolved in our cryo-EM map (Extended Data Fig. 9e, k–n), which enabled de novo modelling of an N-terminal pleckstrin homology domain and a C-terminal LRR domain, which are connected by a flexible linker that stretches perpendicularly across the parental dsDNA (Fig. 3c, d). The pleckstrin homology domain interacts with the ZnF domains of MCM2 and MCM6, parental dsDNA and the N-terminal region of the TIMELESS α-solenoid (Extended Data Fig. 10c, d), consistent with the reported role for TIMELESS–TIPIN in recruiting CUL2[LRR1] to the replisome in *Caenorhabditis elegans*[30]. The LRR domain comprises seven canonical and two irregular LRR motifs and forms a shallow arc, reaching from the parental dsDNA to the N-tier face of MCM3 and MCM5 (Fig. 3d, Extended Data Fig. 10e, f). The BC and CUL2 boxes, which link LRR1 to ELOB–ELOC–CUL2–RBX1, are situated between LRR repeats 8 and 9 (Fig. 3c, d, Extended Data Fig. 10e, f), and a two-stranded antiparallel β-sheet caps the LRR domain at its C-terminal end (Extended Data Fig. 10g). In addition, the C-terminal HMG box of AND-1 could be docked into a small region of density alongside ELOC and LRR1 (Extended data Fig. 10h, i), which was absent in 3D classes that lacked AND-1 (Extended Data Fig. 10j, k), indicating that AND-1 interacts with CUL2[LRR1] in the human replisome.

Remarkably, despite the very different architectures of the LRR1 and Dia2 LRR domains, they bind to the same region of the MCM N-tier, but do so via completely different modes of interaction. The LRR1 LRR domain interacts predominantly with the three-stranded antiparallel β-sheet of the ZnF domain of MCM3, which extends the shallow arc of the LRR1 β-sheet (Extended Data Fig. 10l). This interface is augmented on one side by interactions between the MCM7 N terminus and the tip of the ZnF domain of MCM3 and LRR1 repeats 8 and 9 (Extended Data Fig. 10l, m). On the other side, MCM3 residues 3–8 and 164–174 are significantly rearranged upon CUL2[LRR1] binding, such that the N terminus of MCM3, now projecting between the ZnF domains of MCM3 and MCM5, stabilizes an interaction between MCM3 residues 164–174 and a loop and short helix preceding LRR1 repeat 9 (Extended Data Fig. 10n). Finally, charged residues immediately preceding the β-strands of LRR1 repeats 4–7 form multiple polar contacts with the tip of the ZnF domain of MCM5 (Extended Data Fig. 10m). Further details are illustrated in Extended Data Fig. 10m, n.

## Regulation of CMG ubiquitylation

Ubiquitylation of CMG by both SCF[Dia2] and CUL2[LRR1] is suppressed by the excluded DNA strand at replication forks[8–10]; our discovery that Dia2 and LRR1 bind directly to a common site across the ZnF domains of MCM3 and MCM5 suggested that this region of MCM might be important for the regulation of ubiquitylation. In our recent structure of the human replisome bound to a replication fork[28], cryo-EM density that we attributed to the excluded strand was positioned in the channel between the ZnF domains of MCM3 and MCM5, consistent with previous structures of *Drosophila* and budding yeast CMG[13,31,32]. To further validate our assignment of the excluded strand, we identified a subset of particles lacking CUL2[LRR1] from our dataset of replisomes assembled without an

excluded strand (Extended Data Figs. 8, 9g). In the resulting density map, the MCM N-tier was identical to our previous map of replication fork-associated CMG[28], apart from a single region of density, extending from the fork junction between the ZnF domains of MCM3 and MCM5 (Fig. 4a, Extended Data Fig. 10o). This density was present only in the complex associated with the replication fork, thus confirming that it is contributed by the excluded DNA strand.

Crucially, Fig. 4b shows that the presence of the excluded strand between the ZnF domains of MCM3 and MCM5 sterically blocks the engagement of the Dia2 and LRR1 LRR domains with MCM. As the LRR–MCM interaction is essential for ubiquitylation of CMG and, in turn, replisome disassembly, the occlusion of this interface by the excluded strand provides an elegant and universal explanation for the regulation of replisome disassembly across yeasts and metazoa. Notably, the LRR domains of Dia2 and LRR1 are not demonstrably homologous in sequence or structure. Thus, we propose that the binding of Dia2[LRR] and LRR1[LRR] across the exit channel of the excluded strand reflects convergent evolution, probably indicative of a stringent evolutionary pressure to accurately regulate replisome disassembly, and thereby safeguard replication forks. This evolutionary constraint is not evident in parts of the replisome disassembly machinery that do not contribute to the regulation of CMG ubiquitylation. For example, the Dia2 tetratricopeptide repeat domain binds yeast Mrc1 and Ctf4, whereas the LRR1 pleckstrin homology domain binds human TIMELESS.

On the basis of our results, we propose the model summarized in Fig. 4c. Ubiquitylation of the MCM double hexamer is blocked by the occlusion of the LRR binding site at the inter-hexamer interface[27] (Extended Data Fig. 11a). This occlusion probably also suppresses ubiquitylation during the conversion of MCM double hexamers into pairs of active CMG helicases[33], before the lagging-strand template is excluded. Once bidirectional replication forks are established and elongation begins, the spooling of the excluded DNA strand between the ZnF domains of MCM3 and MCM5 sterically blocks LRR engagement on MCM. It is possible that the binding of proteins to the excluded strand may help to block LRR–MCM engagement. However, ubiquitylation of CMG is inhibited at reconstituted budding yeast replication forks in the absence of the lagging-strand machinery (Extended Data Fig. 11b, c), consistent with the excluded DNA alone being sufficient to suppress SCF[Dia2] during elongation. In principle, the binding of yeast Dia2 to Mrc1 and Ctf4, and human LRR1 to TIMELESS and AND-1, could still occur at replication forks, even when the LRR–MCM interaction is blocked by the excluded strand. Accordingly, Mrc1–Ctf4 can support SCF[Dia2-13A] recruitment to reconstituted replisomes (Extended Data Fig. 11d, e). Critically, however, the essentiality of the LRR–MCM interaction for ubiquitylation of CMG will restrict replisome disassembly to termination, independent of the timing of E3 ligase recruitment, and irrespective of whether a replication fork terminates via fork convergence, or at a telomere.

Finally, we note that if the excluded strand is ever mispositioned, for example, during replication fork stalling or reversal, replisome disassembly could be triggered, due to premature LRR–MCM engagement. As such, the regulatory mechanism that we describe here may have implications for the stability of the replication fork under conditions of replication stress.

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

**Reporting summary**

Further information on research design is available in the Nature Research Reporting Summary linked to this paper.

## Data availability

Cryo-EM density maps of the yeast replisome–SCF$^{Dia2}$ complex on dsDNA have been deposited in the Electron Microscopy Data Bank (EMDB) under the following accession numbers: EMD-13495 (full-complex unsharpened map, conformation I), EMD-13496 (full-complex sharpened map, conformation I), EMD-13497 (multibody refinement (MBR), MCM N-tier, conformation I), EMD-13498 (MBR, MCM C-tier, conformation I), EMD-13500 (full-complex unsharpened map, conformation II), EMD-13512 (full-complex sharpened map, conformation II), EMD-13513 (MBR, MCM N-tier, conformation II), EMD-13514 (MBR, MCM C-tier, conformation II), EMD-13515 (MBR, Dia2–Skp1), EMD-13516 (MBR, Cdc45–GINS–Ctf4–Dpb2$^{NTD}$), EMD-13517 (MBR, Pol-ε$^{non-Cat}$–Mcm5$^{WH}$) and EMD-13518 (full complex enriched for Csm3–Tof1); composite maps produced using Phenix combine_focused_maps have been deposited under accession numbers EMD-13537 (conformation I) and EMD-13539 (conformation II). Cryo-EM density maps of the yeast replisome–SCF$^{Dia2}$ complex in the absence of DNA have been deposited in the EMDB under the following accession numbers: EMD-13519 (full-complex unsharpened map) and EMD-13540 (MBR). Cryo-EM density maps of the human replisome–CUL2$^{LRR1}$ complex used in model building have been deposited in the EMDB under the following accession numbers: EMD-13494 (full complex, consensus refinement), EMD-13491 (MBR, AND-1–CDC45–GINS), EMD-13490 (MBR, ELONGIN–BC–LRR1–CUL2) and EMD-13492 (MBR, CUL2–RBX1). An additional map of the core human replisome not engaged by CUL2$^{LRR1}$ on a DNA substrate lacking a 5′ flap has been deposited under the accession number EMD-13534. Atomic coordinates have been deposited in the Protein Data Bank (PDB) with the accession numbers 7PMK for the yeast replisome–SCF$^{Dia2}$ complex on dsDNA (conformation I), 7PMN for the yeast replisome–SCF$^{Dia2}$ complex on dsDNA (conformation II) and 7PLO for the human replisome–CUL2$^{LRR1}$ complex.

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

**Acknowledgements** We are grateful to R. T. Hay and J. E. Sale for critical feedback on the manuscript; V. Chandrasekharan, T. Dendooven, C. K. Lau and M. E. Wilkinson for practical advice on data processing and model building; S. Chen, G. Sharov, A. Yeates and B. Ahsan for smooth running of the MRC LMB EM facility; J. Grimmett and T. Darling for maintenance of scientific computing facilities; J. Shi for operation of the baculovirus facility; L. Sanchez-Pulido for discussion relating to the evolutionary conservation of Dia2 and LRR1; and C. Polo Rivera and A. Wood for help with flow cytometry analysis. We acknowledge Diamond for access to and support of the cryo-EM facilities at the UK National Electron Bio-Imaging Centre (eBIC), proposals BI23268-20 and BI24557-4, funded by the Wellcome Trust, Medical Research Council (MRC), and Biotechnology and Biological Sciences Research Council (BBSRC). We are grateful to Y. Chaban, O. Raschdorf, S. Masiulis and D. Clare for providing support in the use of the facilities at the eBIC. Ubiquitin and yeast Uba1 were provided by A. Knebel (MRC PPU, Dundee) and antibodies were provided by MRC PPU Reagents and Services (https://mrcppureagents.dundee.ac.uk) unless otherwise stated. This work was supported by the MRC, as part of UK Research and Innovation (MRC grants MC_UP_1201/12 to J.T.P.Y. and MC_UU_12016/13 to K.P.M.L.) and the Wellcome Trust (reference 204678/Z/16/Z for a Sir Henry Wellcome Postdoctoral Fellowship to T.D.D.).

**Author contributions** M.J.-B. performed yeast cryo-EM sample preparation, data acquisition and processing, model building, purification of yeast replisome proteins, created figures, and reviewed and edited the manuscript. M.L.J. performed human cryo-EM sample preparation, data acquisition and processing, model building, created figures, and reviewed and edited the manuscript. Y.B. performed purification of human replisome proteins. K.P.M.L. took part in discussion, acquired funding, and reviewed and edited the manuscript. G.C. acquired the human cryo-EM data. J.T.P.Y. conceptualized and supervised the study, acquired funding, performed yeast biochemistry and optimization of human replisome assembly, purification of yeast and human replisome proteins, wrote the original draft, and reviewed and edited the manuscript. T.D.D. conceptualized and supervised the study, acquired funding, performed yeast biochemistry and genetics, yeast cryo-EM sample preparation, purification of yeast replisome proteins, wrote the original draft, and reviewed and edited the manuscript.

**Competing interests** The authors declare no competing interests.

**Additional information**
**Correspondence and requests for materials** should be addressed to Joseph T. P. Yeeles or Tom D. Deegan.

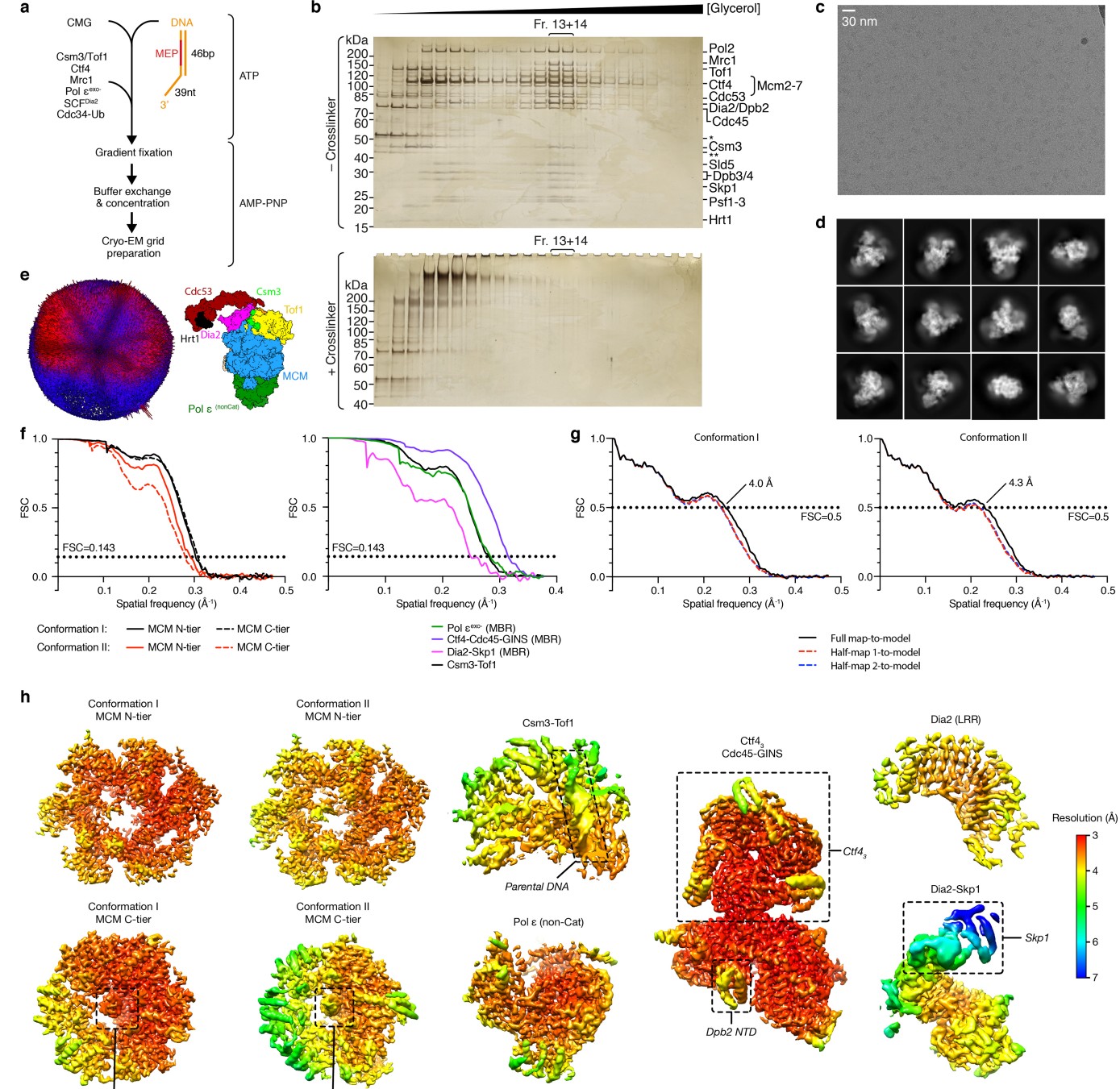

**Extended Data Fig. 1 | Supporting data for cryo-EM investigation of *S. cerevisiae* dsDNA-bound replisome:SCF^Dia2 complexes. a**, Schematic of reconstitution approach used for preparation of cryo-EM sample representing terminating SCF^Dia2-bound replisome complexes after translocation onto dsDNA. A schematic of the DNA substrate used is shown in orange, with the 20 nt tract of methylphosphonate (MEP) linkages coloured red. **b**, Silver-stained SDS-PAGE gels analysing 100 μL fractions taken across 10-30% glycerol gradients, either lacking (top) or containing (bottom) crosslinking agents. Fractions 13+14 used for cryo-EM sample preparation are indicated.

\* = Cdc34-Ub; \*\* = Cdc34. Similar results were observed for three independent sample preparations. **c**, Representative cryo-EM micrograph. **d**, Representative 2D class averages, 40 nm box width. **e**, Representative angular distribution of particle orientations. A correspondingly oriented model is shown to the right for reference. **f**, Fourier shell correlation graphs for maps used in model building. The resolution of reconstructions calculated at the FSC=0.143 criterion are reported in Extended Data Fig. 2. **g**, Model-to-map correlation graphs. **h**, Cryo-EM density maps relevant to model building, coloured by local resolution. For gel source data, see Supplementary Fig. 1.

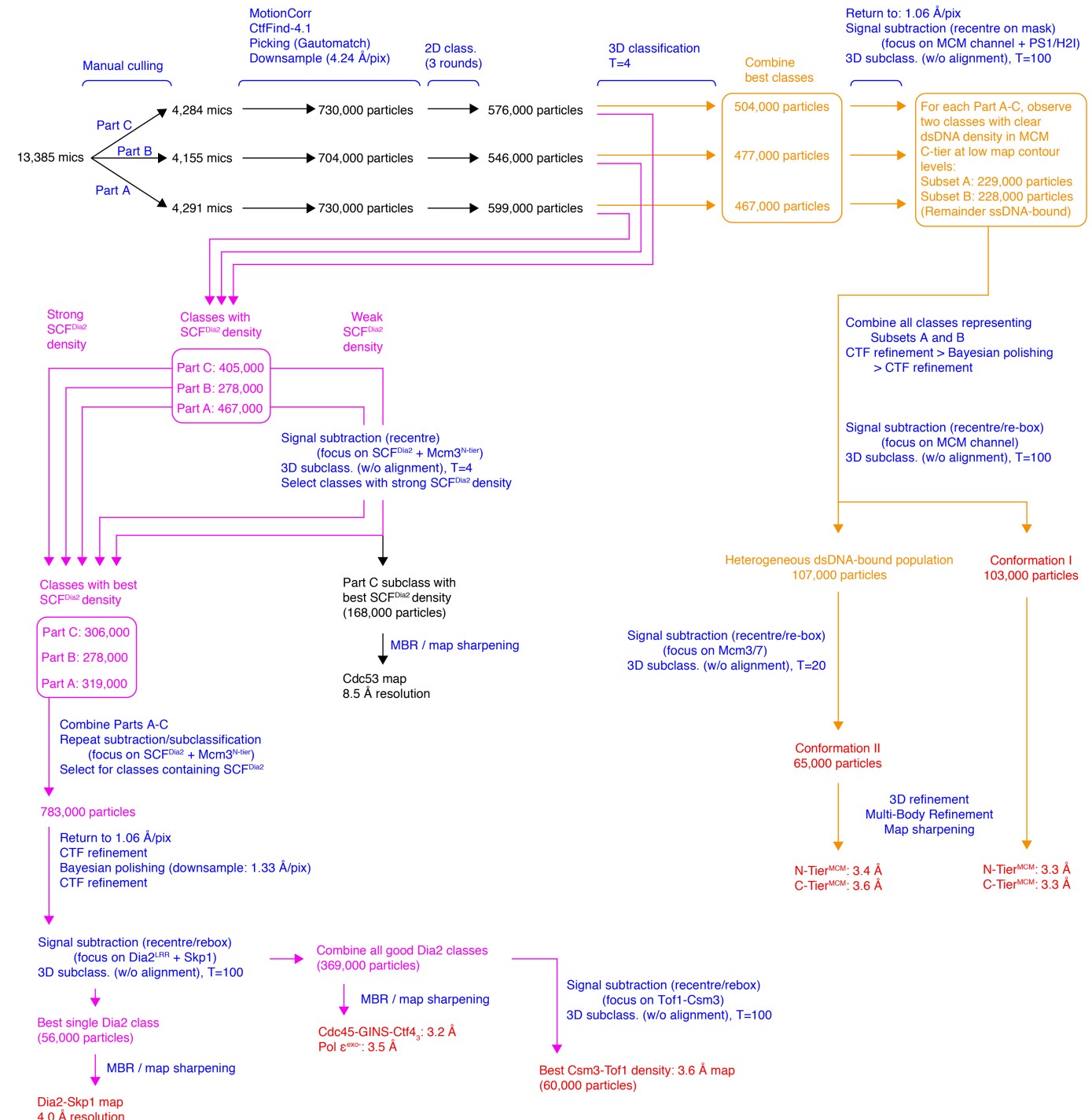

**Extended Data Fig. 2 | Data processing pipeline related to *S. cerevisiae* replisome:SCF$^{Dia2}$ complexes.** The approach used to subclassify complexes based on MCM C-tier conformation and DNA engagement are coloured orange; the approach used to derive cryo-EM reconstructions for model building or adjustment of regions outside MCM are coloured magenta. Reported resolutions are calculated based on the FSC=0.143 criterion (refer to Extended Data Fig. 1f).

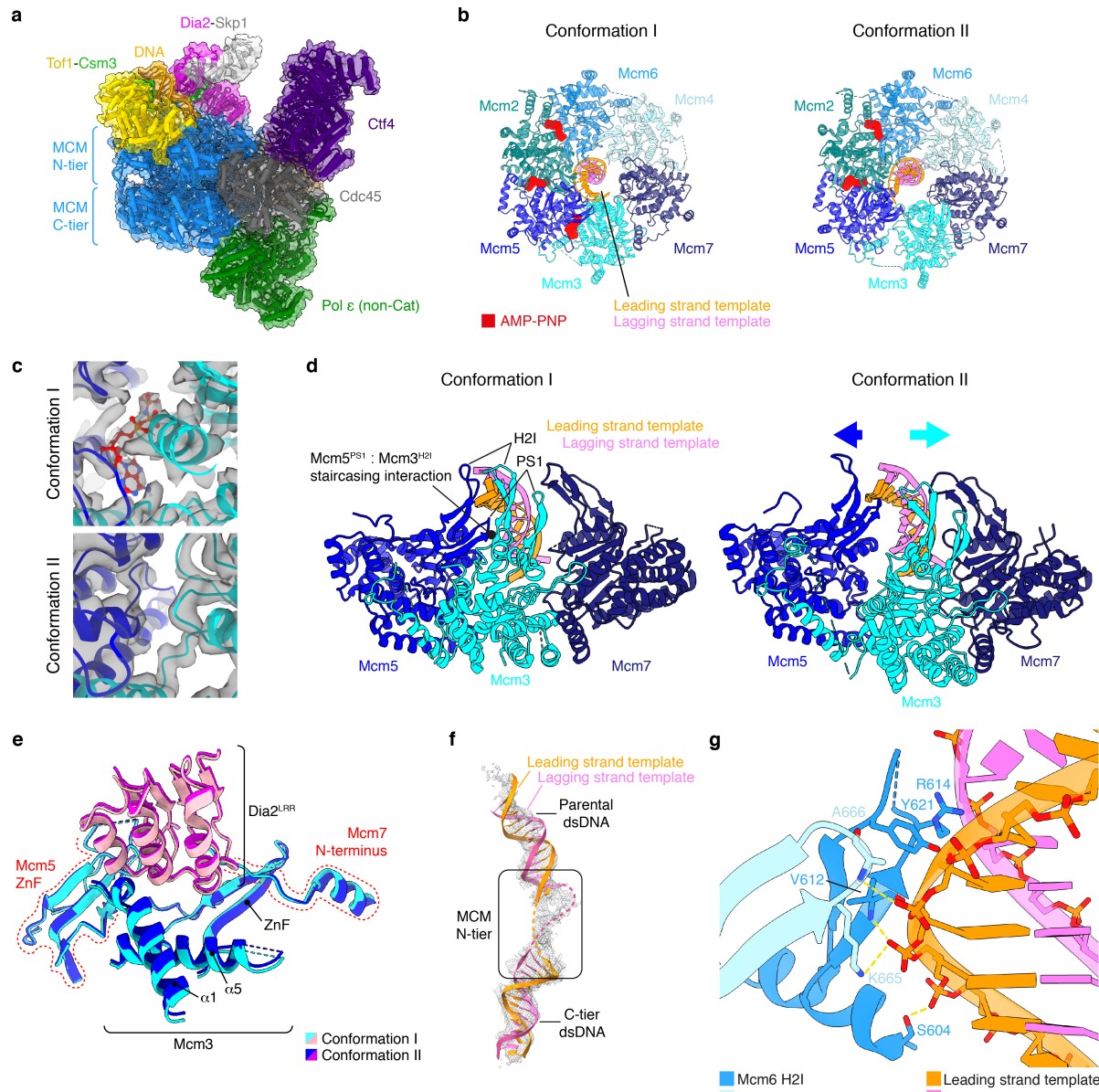

**Extended Data Fig. 3 | DNA engagement by *S. cerevisiae* CMG following translocation onto dsDNA. a**, Comparison of conformations I and II demonstrating the similarity in the overall complex architecture. For clarity, conformation I is rendered as a surface, whilst conformation II is shown as a cartoon. **b**, Overview of MCM C-tier domains bound to dsDNA, highlighting ATPase site occupancy in conformations I and II. **c**, Cryo-EM density (grey) at the MCM C-tier Mcm3-Mcm5 interface for each conformation. Mcm3, cyan; Mcm5, blue; AMP-PNP, red. **d**, Changes in ATPase site occupancy between conformations I and II correspond to movement of Mcm3/5/7 AAA+ domains and their DNA-binding loops (helix-2-insertion, H2I; presensor-1, PS1). Arrows indicate the relative movement of Mcm5 and Mcm3. The outward movement of Mcm3/Mcm7 in conformation II - associated with opening of the Mcm3/5 interface and loss of nucleotide at this ATPase site - leads to loss of the canonical contacts (described in panel **g**) formed between the Mcm3 H2I/PS1

loops and the leading-strand template DNA phosphate backbone. As such conformation II may reflect a partially disengaged state. **e**, Comparison of MCM:Dia2^LRR interface between conformations I and II, demonstrating lack of conformational changes in this region. **f**, Model of DNA in cryo-EM density (mesh) for conformation I demonstrating distortion of the B-form DNA duplex within the MCM N-tier; similar DNA density is observed within the MCM N-tier for conformation II. Approximate trajectories of DNA strands within the MCM N-tier are shown as dotted paths. **g**, Engagement of the leading-strand template DNA phosphate backbone by Mcm H2I/PS1 loops in complexes that have translocated onto dsDNA is comparable to that previously observed for CMG bound to ssDNA[13]. Contacts shown for the representative Mcm6 subunit (conformation I). Specific contacts with the DNA phosphate moieties indicated by dashed yellow lines.

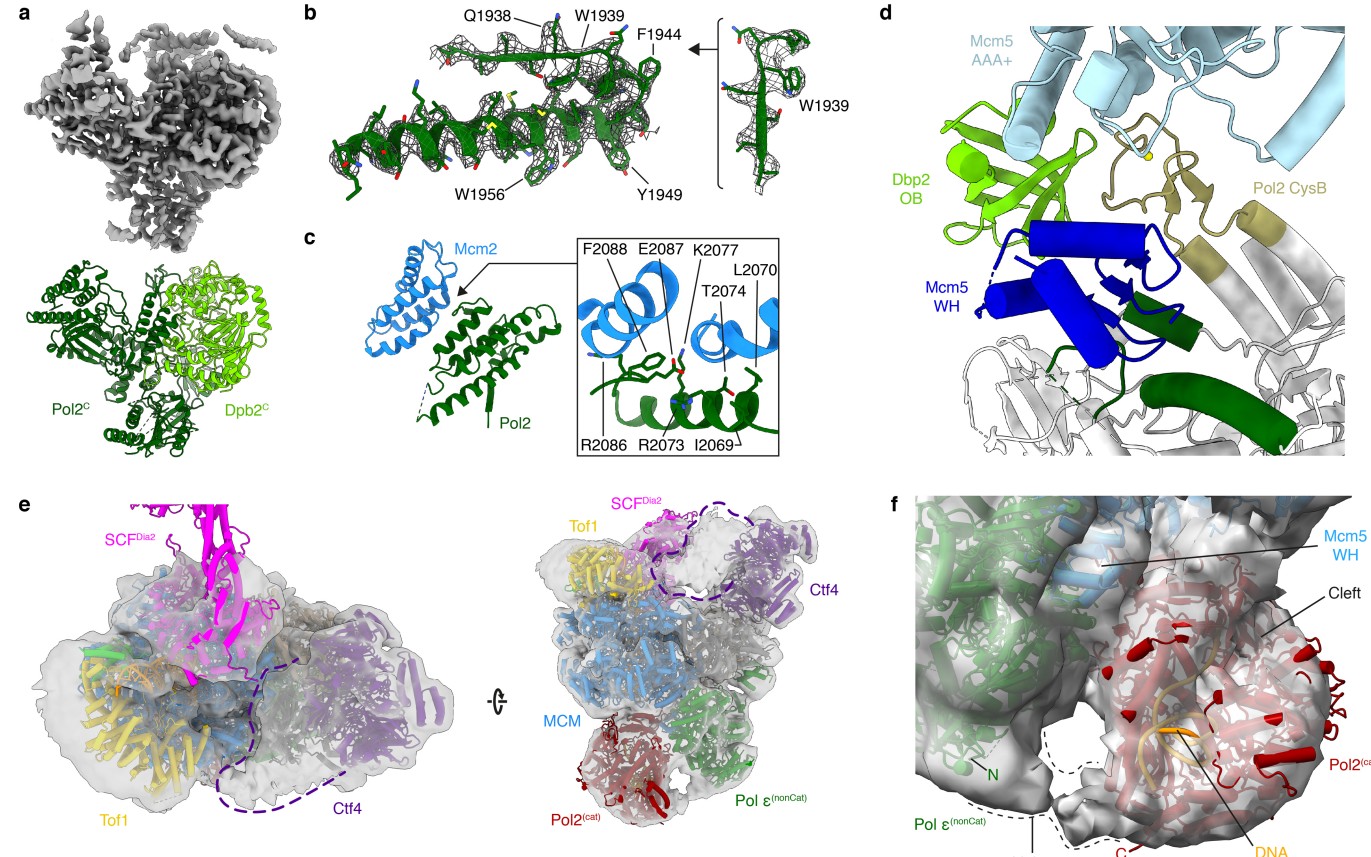

**Extended Data Fig. 4 | Insights into Pol ε positioning within the *S. cerevisiae* replisome. a**, Cryo-EM density derived from multi-body refinement (top) and corresponding atomic model (bottom) of the Pol ε non-catalytic module (Pol ε[non-Cat]), with the exception of the Dpb2 NTD. **b**, Representative cryo-EM density (mesh) allowing de novo model building and adjustment of prior structures. **c**, Pol2-Mcm2 AAA+ domain interface. **d**, Interactions formed by the Mcm5 winged-helix (WH) domain with Pol ε[non-Cat]. The Mcm5 WH is observed to contact regions of Pol2 (dark green) in addition to the Pol2 CysB and Dpb2 OB-fold domains. Interactions depicted in **c** and **d** have not been characterised previously. **e**, Regions of additional cryo-EM density observed for SCF[Dia2]-bound replisome complexes on dsDNA, visible at low map contour levels. The crystal structure of the Pol2 catalytic domain (PDB: 4M8O[34]) has been rigid-body fitted to additional density beside the MCM channel exit. In contrast to previous structures[19,28,35], this positions the Pol2 catalytic domain at the C-tier face of CMG, adjacent to the leading-strand template, in a state that may be important for leading-strand synthesis. Additional unassigned density between Ctf4[SepB], Tof1 and SCF[Dia2] is outlined. **f**, Focused view of additional density attributed to the Pol2 catalytic domain (as in **e**, except rotated 180°). The C-terminal residue of the Pol2 catalytic domain (residue 1186), the N-terminal residue of the Pol2 non-catalytic domain (residue 1321), and density linking the two domains are indicated.

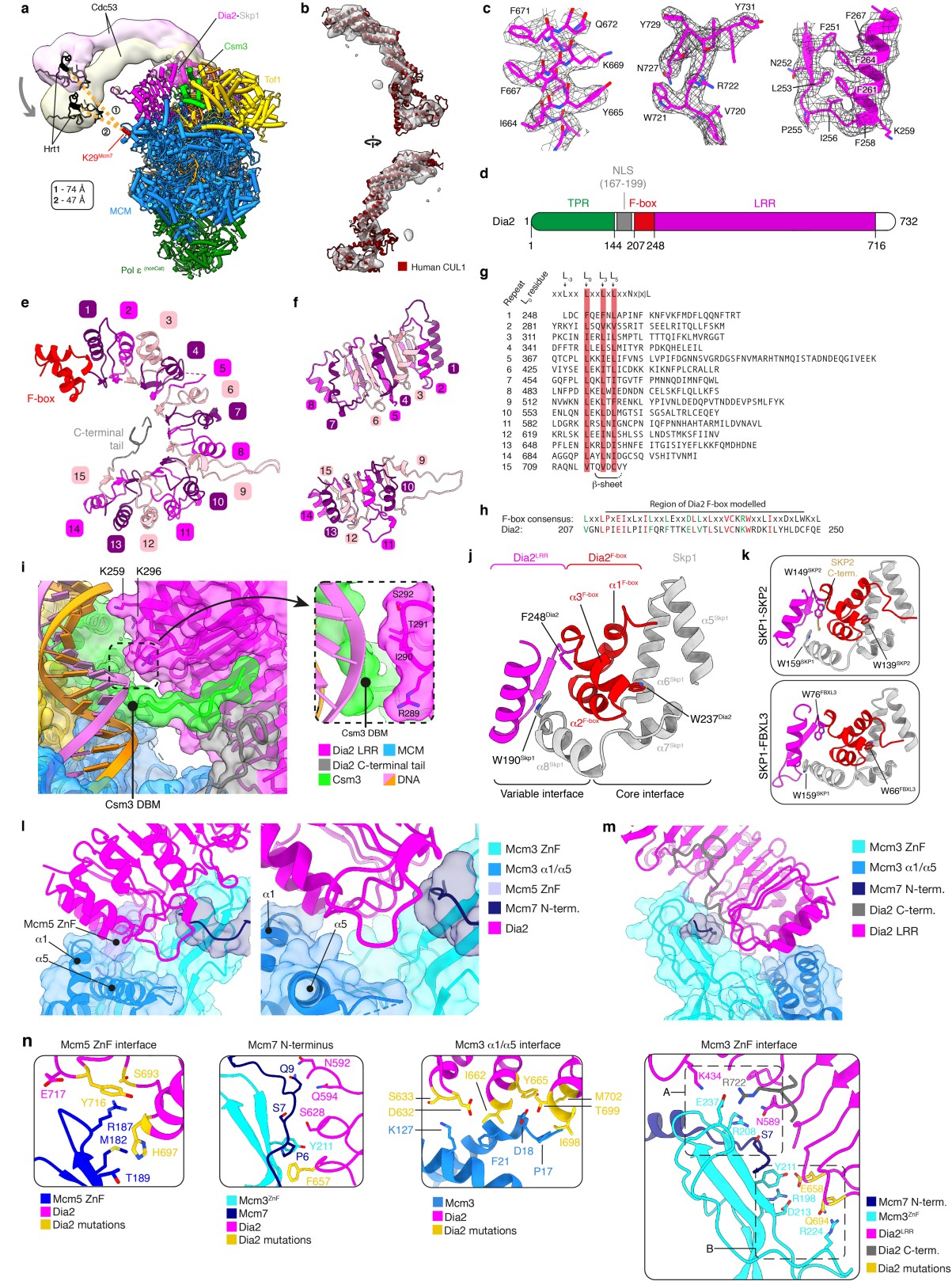

**Extended Data Fig. 5** | See next page for caption.

**Extended Data Fig. 5 | Supporting information for the Dia2 structure and its interaction with the *S. cerevisiae* replisome. a**, Cryo-EM density for Cdc53-Hrt1 for two 3D classes following signal subtraction/3D subclassification, demonstrating the flexibility observed in the position of replisome-bound SCF$^{Dia2}$. The position of Hrt1 is shown, derived from rigid-body fitting the crystal structure of homologous CUL1-RBX1 (PDB: 1LDK[23]). The approximate distance between Hrt1 and the primary ubiquitylation site (K29$^{Mcm7}$) is indicated. **b**, Cryo-EM density map for Cdc53-Hrt1 with the crystal structure of homologous CUL1 (PDB: 1LDK[23]) rigid-body fitted. **c**, Representative cryo-EM density (mesh) across different regions of Dia2. **d**, Dia2 domain architecture; TPR domain and nuclear localisation signal (NLS) as in ref. [24]. **e**, **f**, Alternative views of the Dia2 LRR domain coloured by repeat. The F-box domain and C-terminal tail are shown for context in **e**. **g**, Comparison of Dia2 LRR domain repeats to the LRR consensus sequence[36]. L is Leu/Val/Ile/Phe, N is Asn/Thr/Cys, x is any amino acid; we consider $L_0$ as the first repeat residue. The core LxxLxL motif is highlighted. **h**, Comparison of Dia2 F-box to the consensus sequence[37]. Exact matches coloured red, conservative differences coloured green. **i**, The Dia2 LRR domain (repeats 1 and 2) closely approaches the parental dsDNA (orange: leading-strand template; pink: lagging-strand template) and Csm3. The region of Csm3 upstream of the DNA-binding motif (DBM) is observed to interact with the Dia2 LRR domain β-sheet and Dia2 C-terminal tail, however cryo-EM density for this region was insufficient to identify details of this interaction. Dia2 residues which are positioned close to DNA are labelled. **j**, Dia2-Skp1 interaction. **k**, Interactions of Skp1 with alternative LRR-domain-containing F-box proteins, to show similarity with **j**. Human SKP1-SKP2: PDB:1FQV[37]; human SKP1-FBXL3: PDB: 4I6J[38]. **l**, Overview of interaction between Dia2 LRR domain and Mcm subunits. **m**, The MCM:Dia2$^{LRR}$ interaction does not involve the concave β-sheet surface of Dia2$^{LRR}$. **n**, Detail of the MCM:Dia2$^{LRR}$ interface. Residues altered in Dia2$^{LRR}$ mutants are yellow. For the Mcm3 ZnF, interaction networks are subdivided into those above (A) and below (B) the position of the Mcm7 N-terminus.

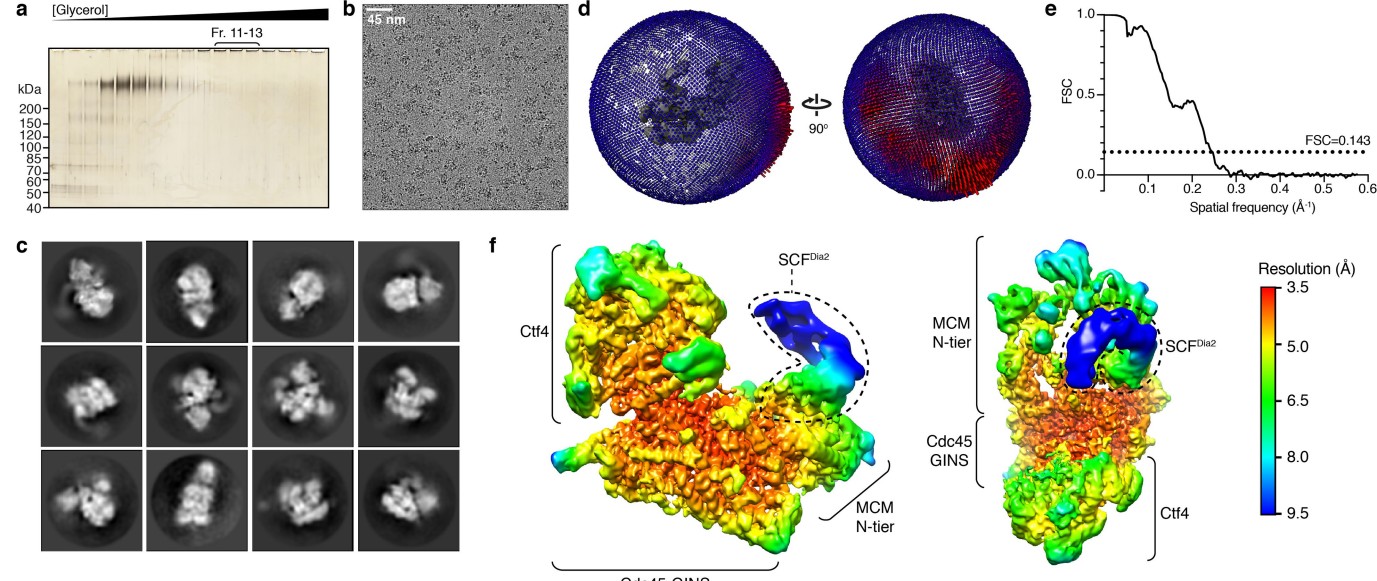

**Extended Data Fig. 6 | Supporting data for cryo-EM investigation of *S. cerevisiae* replisome:SCF^Dia2 complexes assembled in the absence of DNA.**
**a**, Silver-stained SDS-PAGE gel analysing 100 µL fractions taken across a 10-30% GraFix gradient. Fractions 1-17 (of 23) shown. Fractions 11-13 used for cryo-EM sample preparation are indicated. Large-scale sample preparation was performed once; similar results were observed in three independent

small-scale gradient preparations. **b**, Representative cryo-EM micrograph. **c**, Representative 2D class averages, 40 nm box width. **d**, Angular distribution of particle orientations contributing to cryo-EM density map (Fig. 1f). **e**, Fourier shell correlation curve for the multi-body refinement map presented in Fig. 1f. **f**, Cryo-EM density map (related to Fig. 1f) coloured by local resolution. For gel source data, see Supplementary Fig. 1.

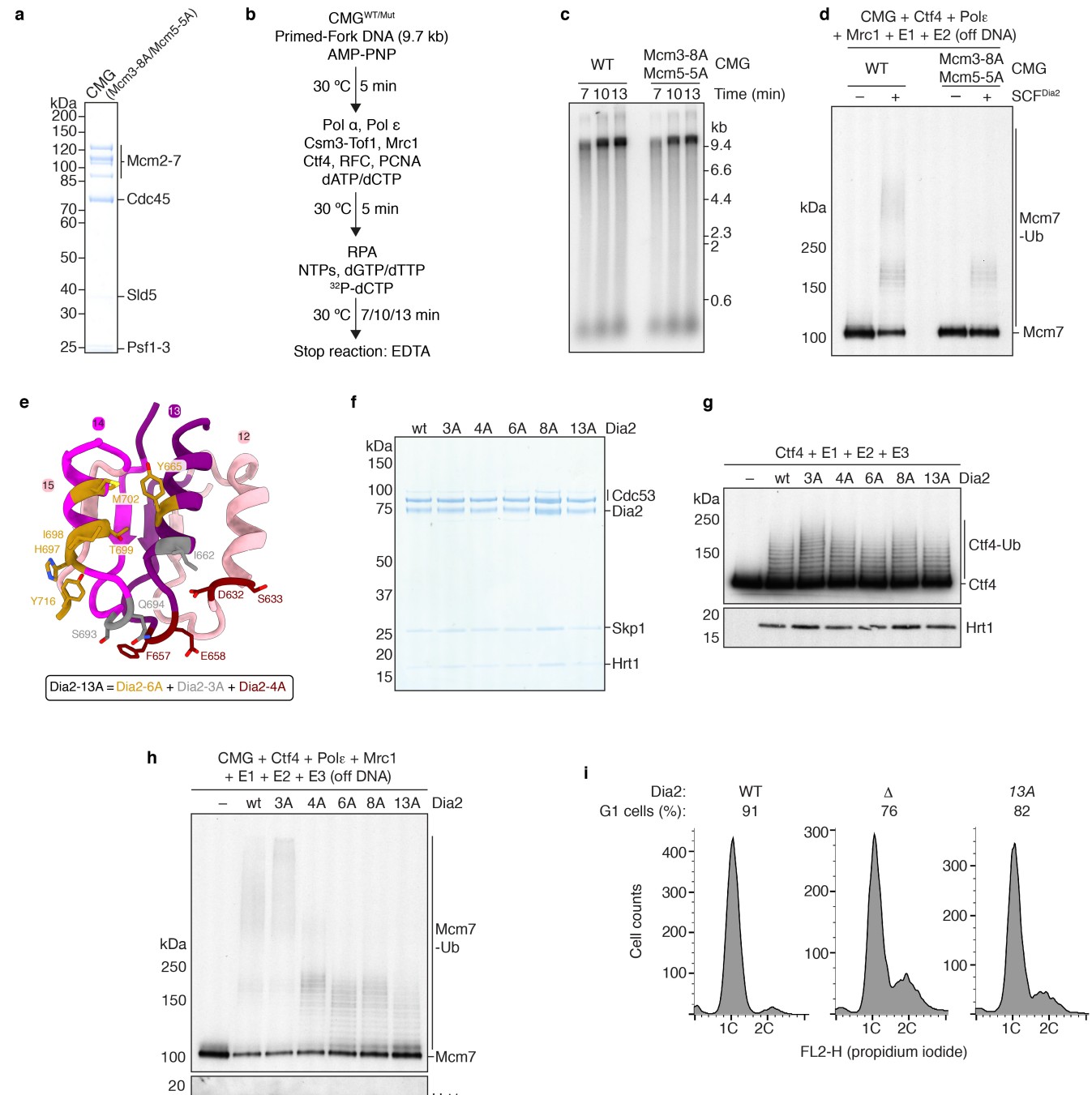

**Extended Data Fig. 7 | Supporting data for functional analyses of Dia2 and MCM mutants. a**, Coomassie-stained SDS-PAGE gel of purified CMG complex containing mutations in Mcm3 and Mcm5 at Dia2[LRR]-MCM interface **b**, Reaction scheme for in vitro replication of 9.7 kb forked DNA template using CMG and the indicated replication proteins. **c**, Reaction conducted as in panel **b** with wildtype or mutant CMG. Samples were separated on an alkaline agarose gel and visualised by auto-radiography. This experiment was performed twice. **d**, In vitro CMG ubiquitylation reaction in the absence of DNA. The indicated proteins were incubated in the presence of ubiquitin and ATP and then visualised by SDS-PAGE and immunoblotting. This experiment was repeated three times. **e**, Positions of residues mutated in Dia2 LRR domain. LRR repeats 12-15 are coloured and numbered as in Extended Data Fig. 5e, f. Residues are coloured according to the Dia2 mutant in which they are present; all residues

shown were mutated in Dia2-13A. Dia2-8A featured the following mutations: D632A, F657A, I662A, Y665A, Q694A, I698A, T699A and Y716A. **f**, Coomassie-stained SDS-PAGE gel of purified SCF[Dia2] complexes containing Dia2[LRR] mutants **g**, In vitro Ctf4 ubiquitylation reaction. The indicated proteins were incubated in the presence of ubiquitin and ATP and then visualised by SDS-PAGE and immunoblotting. The Hrt1 immunoblot serves as a loading control for SCF[Dia2]. This experiment was repeated twice. **h**, Reaction conducted as in panel **d** with the indicated Dia2[LRR] mutants. This experiment was repeated three times. **i**, DNA content of G1-arrested cells from experiment in Fig. 2c was monitored by flow cytometry after propidium iodide staining. The proportion of G1 cells, expressed as a percentage of the total cells, is given. For details of gating strategy and assignment of the G1 peak see Supplementary Fig. 2. For gel source data, see Supplementary Fig. 1.

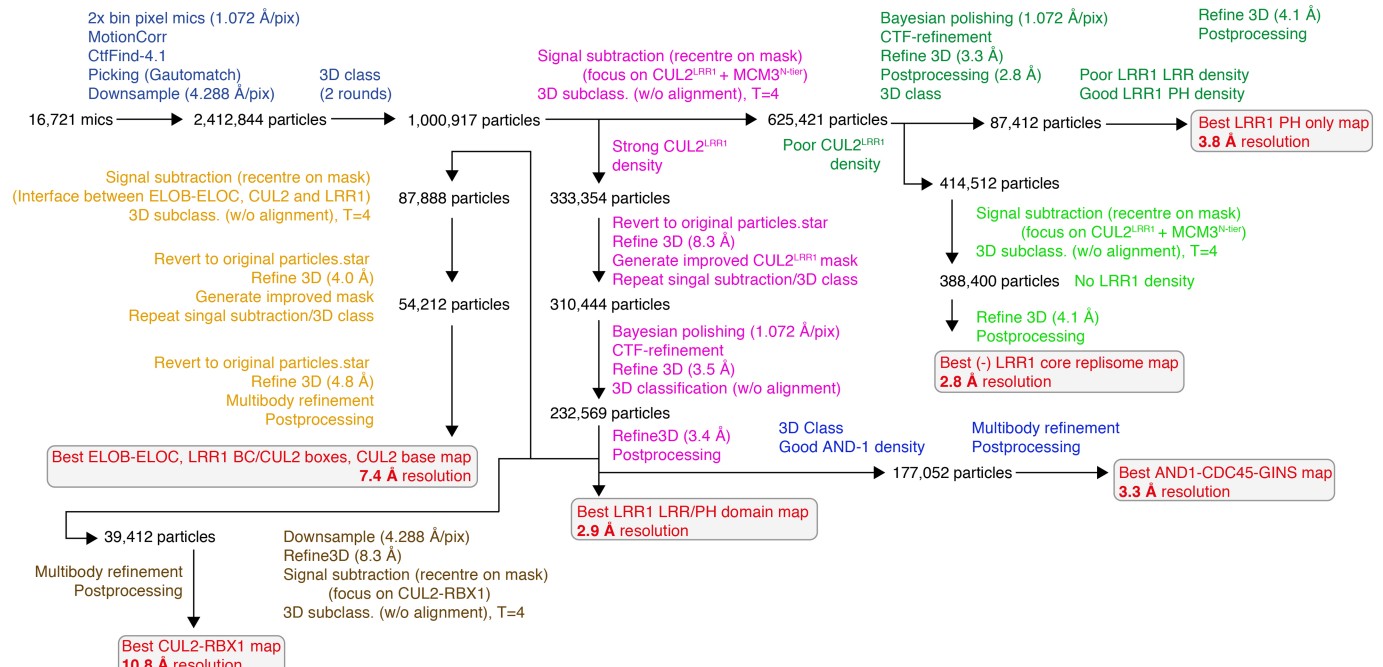

**Extended Data Fig. 8 | Data processing pipeline related to *H. sapiens* replisome:CUL2^LRR1 complexes.** Each pathway describing the generation of a discrete reconstruction is given its own colour. The final reconstructions are coloured red and boxed.

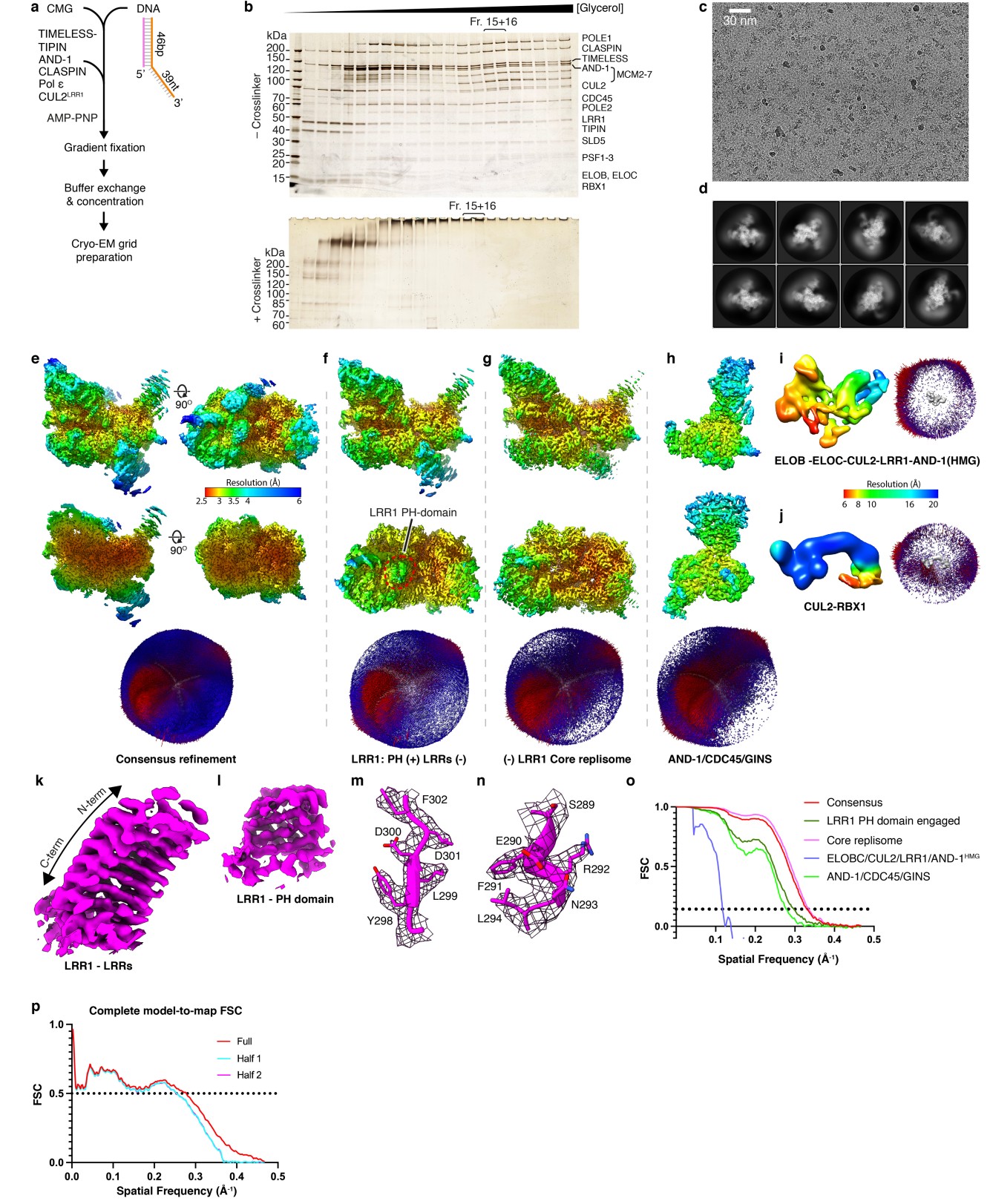

**Extended Data Fig. 9** | See next page for caption.

**Extended Data Fig. 9 | Supporting data for cryo-EM investigation of *H. sapiens* replisome: CUL2<sup>LRR1</sup> complexes. a**, Schematic of reconstitution approach used for preparation of replisomes bound to CUL2<sup>LRR1</sup> for cryo-EM. A schematic of the DNA substrate used is shown with a 39 nucleotide 3′ arm and no 5′ arm. **b**, Silver-stained SDS-PAGE gels analysing 100 μL fractions taken across 10-30% glycerol gradients, either lacking (top) or containing (bottom) crosslinking agents. Fractions 15+16 used for cryo-EM sample preparation are indicated. This experiment was performed twice. **c**, Representative cryo-EM micrograph. **d**, Representative 2D class averages, 40 nm box width. **e**–**j**, (Top) cryo-EM reconstructions coloured by local resolution according to inset keys (Bottom) angular distribution of particle orientations. **e**, Consensus refinement for replisome:CUL2<sup>LRR1</sup> fully engaged. **f**, Consensus refinement for replisome:CUL2<sup>LRR1</sup> where the LRR1<sup>PH</sup> domain is bound but the LRRs are disengaged. **g**, Consensus refinement for particles lacking CUL2<sup>LRR1</sup>. **h**, Multi-body refinement for AND-1:CDC45:GINS. **i**, Multi-body refinement for LRR1:ELOB:ELOC:CUL2:AND-1-HMG. **j**, Multi-body refinement for CUL2:RBX1. **k**, Cryo-EM density for the LRR1 LRRs. **l**, Cryo-EM density for the LRR1 PH domain. **m**, Representative cryo-EM density for a LRR1 LRR domain β-strand at 3.5 Å resolution. **n**, Representative cryo-EM density for a LRR1 LRR domain α-helix at 3.7 Å resolution. **o**, Fourier-shell correlation (FSC) curves for the various maps used in model building. **p**, Map-to-model FSC curves for the complete model docked into the consensus refinement for replisomes fully engaged by CUL2<sup>LRR1</sup>. For gel source data, see Supplementary Fig. 1.

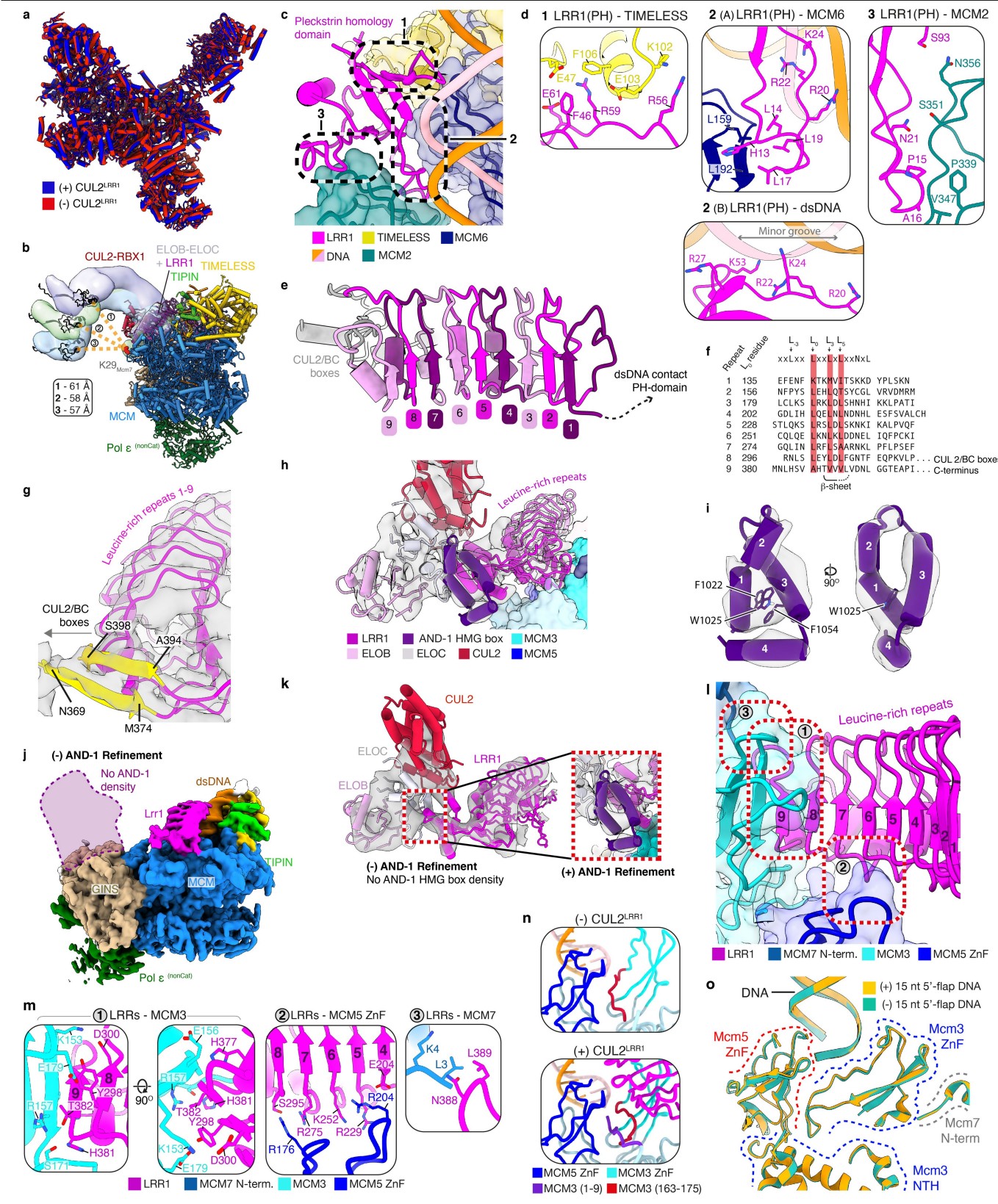

**Extended Data Fig. 10** | See next page for caption.

**Extended Data Fig. 10 | Supporting information for the CUL2$^{LRR1}$ structure and its interaction with the *H. sapiens* replisome. a**, Structural overlay of aligned model from replisomes bound to CUL2$^{LRR1}$ (blue) and in the absence of CUL2$^{LRR1}$ (red). **b**, Composite model and map representing the conformational variability of CUL2/RBX1. The model for the replisome, bound to LRR1 and ELOB-ELOC, is displayed using pipes and planks rendering and coloured according to subunit. Three representative 3D classes are displayed encompassing density for CUL2:RBX1 obtained through 3D classification without alignment. The distance between RBX1 and K29$_{MCM7}$ is indicated as a dotted orange line and distances denoted in the inset key. **c**, Overview of the interface between the LRR1 PH domain and the replisome. Subunits interacting with the LRR1 PH domain are displayed using transparent surface rendering. Boxed regions indicate key interaction interfaces expanded in panel **d**. **d**, Detailed structural views of the interface between the LRR1 PH domain and 1: TIMELESS, 2(A): MCM6 ZnF, 2(B): dsDNA and 3: MCM2 ZnF. **e**, Model for the LRR1 LRRs with numbering indicating the order of the leucine-rich repeats. **f**, Consensus motif for the LRR1 LRRs. The sequence of each repeat is indicated with the positions of the key $L_0$, $L_3$ and $L_5$ residues highlighted in red. Repeats 1 and 9 represent irregular LRRs. L is Leu/Val/Ile/Phe, N is Asn/Thr/Cys, x is any amino acidL is Leu/Val/Ile/Phe, N is Asn/Thr/Cys, x is any amino acid. **g**, LRR1 model docked into transparent cryo-EM density with the capping 2-stranded β-sheet highlighted in gold. **h**, Overview of the LRR1:ELOB:ELOC:CUL2:AND-1 interface. Models displayed docked into transparent cryo-EM density with MCM subunits visualised using surface rendering. **i**, Structure of the AND-1 HMG box (PDB:2D7L) docked into the AND-1-dependent cryo-EM density adjacent to ELOC and LRR1. Selected hydrophobic core residues displayed. **j**, Map of the replisome bound to CUL2$^{LRR1}$ in the absence of AND-1 coloured according to subunit. **k**, Cryo-EM density of the LRR1:ELOB:ELOC:CUL2 interface obtained through multi-body refinement from particles lacking AND-1. The density attributed to the AND-1 HMG box is dependent upon AND-1. **l**, Overview of the MCM:LRR1$^{LRR}$ interface. MCM subunits displayed with additional transparent surface rendering and the order of the LRR1$^{LRRs}$ numbered. Red-dashed boxes indicate key interaction sites, expanded in panel **m**. **m**, Detail of the MCM:LRR1$^{LRR}$ interface involving contacts between the LRR1$^{LRRs}$ and 1 - MCM3, 2 - MCM5 ZnF and 3 - the MCM7 N-terminus. **n**, Model highlighting local rearrangements of MCM3 upon binding CUL2$^{LRR1}$. Structures in the absence (top) and presence (bottom) of CUL2$^{LRR1}$, coloured according to inset key, highlight the rearrangement of MCM3$_{(1-9)}$ and MCM3$_{(164-174)}$. **o**, Comparison of the CUL2$^{LRR1}$-interacting regions of MCM from complexes assembled on a DNA substrate either lacking a 5'-flap (green) or containing a 15 nucleotide 5'-flap (gold, PDB: 7PFO[28]). Complexes lacked CUL2$^{LRR1}$. The r.m.s.d. between the two structures for the region shown is 0.43 Å.

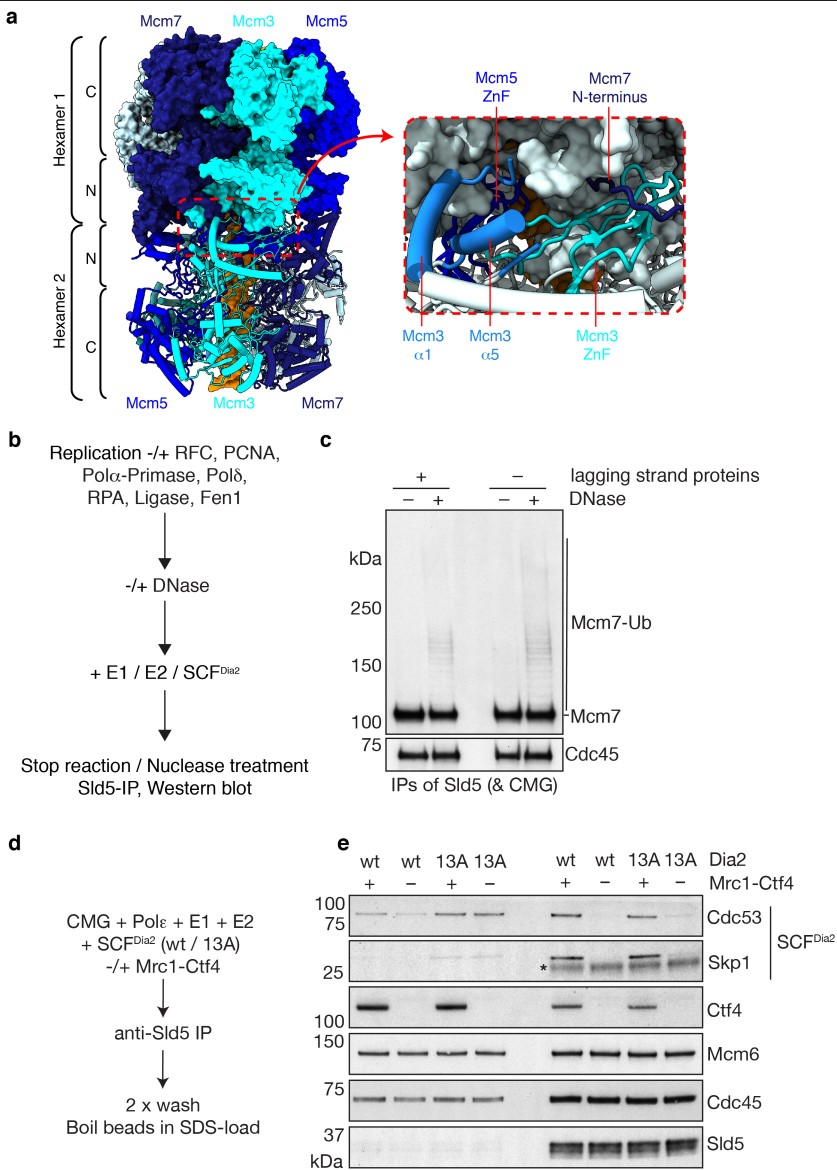

**Extended Data Fig. 11 | Supporting data for model for regulation of CMG ubiquitylation. a**, The MCM:Dia2[LRR] interface is occluded in the inactive Mcm2-7 double hexamer. The structure of the budding yeast Mcm2-7 double hexamer is shown (PDB: 5BK4 (ref. [39])): one Mcm2-7 hexamer is displayed as a cartoon, the other as a surface. Double-stranded DNA is coloured orange. The positions of the N-tier (N) and C-tier (C) are labelled for each hexamer. Inset: focused view of the regions of Mcm2-7 involved in interaction with the Dia2 LRR domain, demonstrating the inaccessibility of these regions to Dia2 in the context of a double hexamer. **b**, Reaction scheme for experiment in panel **c**, to monitor the suppression of CMG ubiquitylation by DNA in the absence of the indicated proteins (top), which are predicted to interact with the excluded DNA strand during lagging strand synthesis. Pif1 was omitted to block fork convergence. DNase was included after the replication step to release the replisome from DNA, which triggers CMG ubiquitylation[8]. **c**, Reaction conducted as in panel **b** and analysed by SDS-PAGE and immunoblot. This experiment was repeated twice. **d**, Reaction scheme for experiment in panel **e**, to monitor the interaction of SCF[Dia2] with the replisome. **e**, Reaction conducted as in panel **d** and analysed by SDS-PAGE and immunoblot. * is rabbit IgG. This experiment was repeated twice. For gel source data, see Supplementary Fig. 1.

**Extended Data Table 1 | Cryo-EM statistics**

| | *S. cerevisiae* Replisome-SCF<sup>Dia2</sup> conformation I (EMD-13537) † (PDB: 7PMK) | *S. cerevisiae* Replisome-SCF<sup>Dia2</sup> conformation II (EMD-13539) † (PDB: 7PMN) | *H. sapiens* Replisome-CUL2<sup>LRR1</sup> (EMD-13494) † (PDB: 7PLO) |
|---|---|---|---|
| **Data collection and processing** | | | |
| Magnification | 81,000 X | 81,000 X | 81,000 X |
| Voltage (kV) | 300 | 300 | 300 |
| Electron exposure ($e^-/Å^2$) | 38.8 | 38.8 | 38.3 |
| Defocus range (μm) | -0.4 to -2.2 | -0.4 to -2.2 | -0.8 to -2.8 |
| Pixel size (Å) [super-resolution] | 0.53 | 0.53 | 0.536 |
| Symmetry imposed | None | None | None |
| Initial particle images (no.) | 2,160,000 | 2,160,000 | 2,412,000 |
| Final particle images (no.) † | 56,000 – 369,000 | 56,000 – 369,000 | 39,000 – 232,000 |
| Map resolution (Å) † 0.143 FSC threshold | 3.2 – 4.0 | 3.2 – 4.0 | 2.8 – 10.8 |
| Map resolution range (Å) | 3 – 7 | 3 – 7 | 3 – 20 |
| | | | |
| **Refinement** | | | |
| Initial model used (PDB code) | 6SKL,6HV9,1NEX | 6SKL,6HV9,1NEX | 6XTX, 5MQI |
| Model resolution (Å) 0.5 FSC threshold | 4.0 | 4.3 | 3.2 |
| Map sharpening $B$ factor ($Å^2$) † | -20 to -50 | -20 to -50 | -30 to -100 |
| Model composition | | | |
| Non-hydrogen atoms | 74,708 | 74,666 | 78,016 |
| Protein residues | 9,130 | 9,134 | 9754 |
| Ligands | 7 $Zn^{2+}$, 3 $Mg^{2+}$, 3 AMP-PNP | 7 $Zn^{2+}$, 2 $Mg^{2+}$, 2 AMP-PNP | 7 $Zn^{2+}$, 3 $Mg^{2+}$, 3 AMP-PNP |
| $B$ factors ($Å^2$) | | | |
| Protein | 112.76 | 99.89 | 66.61 |
| Ligand | 109.20 | 80.56 | 40.51 |
| R.m.s. deviations | | | |
| Bond lengths (Å) | 0.006 | 0.006 | 0.005 |
| Bond angles (°) | 0.930 | 0.930 | 0.890 |
| Validation | | | |
| MolProbity score | 0.74 | 0.78 | 1.03 |
| Clashscore | 0.37 | 0.32 | 0.53 |
| Poor rotamers (%) | 0.31 | 0.53 | 1.01 |
| Ramachandran plot | | | |
| Favored (%) | 97.45 | 97.12 | 95.09 |
| Allowed (%) | 2.55 | 2.88 | 4.81 |
| Disallowed (%) | 0.00 | 0.00 | 0.10 |

Model statistics generated using Phenix comprehensive validation (cryo-EM)[40]. †, refer to Extended Data Figs. 2, 8 and Methods for details related to individual maps.

# Reporting Summary

## Statistics

For all statistical analyses, confirm that the following items are present in the figure legend, table legend, main text, or Methods section.

| n/a | Confirmed | |
|---|---|---|
| ☒ | ☐ | The exact sample size (*n*) for each experimental group/condition, given as a discrete number and unit of measurement |
| ☐ | ☒ | A statement on whether measurements were taken from distinct samples or whether the same sample was measured repeatedly |
| ☒ | ☐ | The statistical test(s) used AND whether they are one- or two-sided<br>*Only common tests should be described solely by name; describe more complex techniques in the Methods section.* |
| ☒ | ☐ | A description of all covariates tested |
| ☒ | ☐ | A description of any assumptions or corrections, such as tests of normality and adjustment for multiple comparisons |
| ☒ | ☐ | A full description of the statistical parameters including central tendency (e.g. means) or other basic estimates (e.g. regression coefficient) AND variation (e.g. standard deviation) or associated estimates of uncertainty (e.g. confidence intervals) |
| ☒ | ☐ | For null hypothesis testing, the test statistic (e.g. *F*, *t*, *r*) with confidence intervals, effect sizes, degrees of freedom and *P* value noted<br>*Give P values as exact values whenever suitable.* |
| ☒ | ☐ | For Bayesian analysis, information on the choice of priors and Markov chain Monte Carlo settings |
| ☒ | ☐ | For hierarchical and complex designs, identification of the appropriate level for tests and full reporting of outcomes |
| ☒ | ☐ | Estimates of effect sizes (e.g. Cohen's *d*, Pearson's *r*), indicating how they were calculated |

*Our web collection on statistics for biologists contains articles on many of the points above.*

## Software and code

Policy information about availability of computer code

| Data collection | EPU-2 (Thermo-Fisher Scientific) |
|---|---|
| Data analysis | RELION-3.0 and -3.1; CryoSPARC-2; Gautomatch-0.53 and -0.56; MotionCor-2; CTFFIND-4.1; Gctf-1.18; EMAN-2; Xmipp-3; ISOLDE-1.0.1; COOT-0.9.5; UCSF ChimeraX-1.0; UCSF Chimera-1.8.1; Phenix-1.19; Prism-8.0.0; FlowJo-10.8; PyMOL-2.4.1 |

For manuscripts utilizing custom algorithms or software that are central to the research but not yet described in published literature, software must be made available to editors and reviewers. We strongly encourage code deposition in a community repository (e.g. GitHub). See the Nature Portfolio guidelines for submitting code & software for further information.

## Data

Policy information about availability of data

All manuscripts must include a data availability statement. This statement should provide the following information, where applicable:

- Accession codes, unique identifiers, or web links for publicly available datasets
- A description of any restrictions on data availability
- For clinical datasets or third party data, please ensure that the statement adheres to our policy

Cryo-EM density maps of the yeast replisome-SCFDia2 complex on dsDNA have been deposited in the Electron Microscopy Data Bank (EMDB) under the following accession numbers: EMD-13495 (full complex unsharpened map, conformation I); EMD-13496 (full complex sharpened map, conformation I); EMD-13497 (multi-body refinement [MBR], MCM N-tier, conformation I); EMD-13498 (MBR, MCM C-tier, conformation I); EMD-13500 (full complex unsharpened map, conformation II); EMD-13512 (full complex sharpened map, conformation II); EMD-13513 (MBR, MCM N-tier, conformation II); EMD-13514 (MBR, MCM C-tier, conformation II); EMD-13515 (MBR, Dia2-Skp1); EMD-13516 (MBR, Cdc45-GINS-Ctf4-Dpb2NTD); EMD-13517 (MBR, Pol enon-Cat-Mcm5WH); EMD-13518 (full complex enriched for Csm3-Tof1); composite maps produced using Phenix combine_focused_maps have been deposited under accession numbers EMD-13537 (conformation I) and

EMD-13539 (conformation II). Cryo-EM density maps of the yeast replisome-SCFDia2 complex in the absence of DNA have been deposited in the EMDB under the following accession numbers: EMD-13519 (full complex unsharpened map) and EMD-13540 (MBR). Cryo-EM density maps of the human replisome-CUL2LRR1 complex used in model building have been deposited in the EMDB under the following accession numbers: EMD-13494 (full complex, consensus refinement), EMD-13491 (MBR, AND-1/CDC45/GINS), EMD-13490 (MBR, ELONGIN-BC/LRR1/CUL2), EMD-13492 (MBR, CUL2-RBX1). An additional map of the core human replisome not engaged by CUL2LRR1 on a DNA substrate lacking a 5'-flap has been deposited under the accession number EMD-13534. Atomic coordinates have been deposited in the Protein Data Bank (PDB) with the accession numbers 7PMK for the yeast replisome-SCFDia2 complex on dsDNA (conformation I), 7PMN for the yeast replisome-SCFDia2 complex on dsDNA (conformation II), and 7PLO for the human replisome-CUL2LRR1 complex.

# Field-specific reporting

Please select the one below that is the best fit for your research. If you are not sure, read the appropriate sections before making your selection.

☒ Life sciences ☐ Behavioural & social sciences ☐ Ecological, evolutionary & environmental sciences

For a reference copy of the document with all sections, see nature.com/documents/nr-reporting-summary-flat.pdf

# Life sciences study design

All studies must disclose on these points even when the disclosure is negative.

| | |
|---|---|
| Sample size | Our study does not include cohort/population based analysis or comparison and thus does not entail predetermination of sample size. Cryo-EM data were collected, as described in methods; these numbers of micrographs were sufficient to either allow model building or comparative analysis. |
| Data exclusions | During processing of cryo-EM data, poor quality micrographs/particles were excluded based on manual inspection and 2D/3D classification. |
| Replication | Cryo-EM datasets for yeast and human complexes comprised individual sample preparations and datasets. Complex formation was found to be reproducible across multiple independent sample preparations. Details of the number of experimental repeats have been acknowledged in the relevant figure legends. All attempts at data replication were successful. |
| Randomization | For calculation of the resolution of the cryo-EM reconstructions, Fourier shell correlations were calculated using independent halves of the complete datasets, into which the component particles were segregated randomly. |
| Blinding | Our analysis did not require blinding because it did not involve human subjects or live animals. |

# Reporting for specific materials, systems and methods

We require information from authors about some types of materials, experimental systems and methods used in many studies. Here, indicate whether each material, system or method listed is relevant to your study. If you are not sure if a list item applies to your research, read the appropriate section before selecting a response.

## Materials & experimental systems

| n/a | Involved in the study |
|---|---|
| ☐ | ☒ Antibodies |
| ☐ | ☒ Eukaryotic cell lines |
| ☒ | ☐ Palaeontology and archaeology |
| ☒ | ☐ Animals and other organisms |
| ☒ | ☐ Human research participants |
| ☒ | ☐ Clinical data |
| ☒ | ☐ Dual use research of concern |

## Methods

| n/a | Involved in the study |
|---|---|
| ☒ | ☐ ChIP-seq |
| ☐ | ☒ Flow cytometry |
| ☒ | ☐ MRI-based neuroimaging |

# Antibodies

| | |
|---|---|
| Antibodies used | All polyclonal primary antibodies used in this study were raised in sheep against the indicated S. cerevisiae proteins by the Labib laboratory in conjunction with MRC PPU Reagents and Services (https://mrcppureagents.dundee.ac.uk/). Mcm5 (160), Mcm6 (161), Mcm7 (19), Cdc45 (29), Psf1 (58), Sld5 (32), Ctf4 (30), Hrt1 (203). Peroxidase anti-peorxidase (Sigma, P1291) conjugate to horseradish peroxidase of anti-sheep IgG from donkey (Sigma, A3415). |
| Validation | With the exception of anti-Hrt1 (203), the validation of all antibodies is described in Gambus et al, Nat. Cell Biol., 2006 (doi: 10.1038/ncb1382). Anti-Hrt1 (203) was validated by immunoblotting against purified SCFDia2 and yeast whole cell extracts containing wildtype Hrt1 or 6HA-Hrt1. |

# Eukaryotic cell lines

| | |
|---|---|
| Cell line source(s) | SF9 cells were obtained from OXFORD EXPRESSION TECHNOLOGIES, LTD, Cat No. 600100. High-5 cells (bti-tn-5b1-4) were obtained from Thermo-Scientific Cat no. B85502. |
| Authentication | Cell lines were not authenticated. |
| Mycoplasma contamination | Cell lines tested negative for mycoplasma contamination. |
| Commonly misidentified lines (See ICLAC register) | None |

# Flow Cytometry

## Plots

Confirm that:

☒ The axis labels state the marker and fluorochrome used (e.g. CD4-FITC).

☒ The axis scales are clearly visible. Include numbers along axes only for bottom left plot of group (a 'group' is an analysis of identical markers).

☒ All plots are contour plots with outliers or pseudocolor plots.

☒ A numerical value for number of cells or percentage (with statistics) is provided.

## Methodology

| | |
|---|---|
| Sample preparation | For each sample, 10 million yeast cells were harvested and fixed by resuspension in 1 ml of 70% ethanol. Subsequently, 3 ml of 50 mM sodium acetate and 50mg of RNase A was added to 150 uL of fixed cells, followed by incubation at 37°C for 2 h. The cells were then pelleted and proteins degraded by incubation at 37°C for 30 min in 500 uL of 50 mM HCl containing 2.5 mg of Pepsin. Finally, cells were pelleted and then re-suspended in 1 ml of 50 mM sodium citrate containing 2 mg of propidium iodide. Samples were sonicated and then analysed. |
| Instrument | FACSCanto II flow cytometer (Becton Dickinson) |
| Software | FlowJoTM v10.8 software Software (TreeStar Inc.) |
| Cell population abundance | The percentage of cells in G1-phase is provided in EDF 7i. |
| Gating strategy | Gating was performed to remove cell debris and dead cells based on forwards and side scatter properties, as exemplified in Supplementary Fig. 2 |

☒ Tick this box to confirm that a figure exemplifying the gating strategy is provided in the Supplementary Information.

