## [Peer Review File · Nature]

Manuscript Title: A Conserved Mechanism for Regulating Replisome Disassembly in Eukaryotes

Editorial Notes:

Redactions – unpublished data

Reviewer Comments & Author Rebuttals

Reviewer Reports on the Initial Version:

Referee #1:

The manuscript of Jenkyn-Bedford and Jones et al. represents a tour-de-force characterisation of the mechanism of replisome disassembly in eukaryotes. Several structures of yeast and human replisome-E3 assemblies are presented that reveal the interactions and associations necessary for replisome disassembly and CMG ubiquitylation. Biochemical reconstitutions revealed loss-of-function mutants in Dia2 that fail to stimulate MCM7 ubiquitylation, leading to accumulation of intact CMG in G1 due to failure of disassembly. I applaud the authors for an outstanding work. The extent of the biochemical reconstitutions and careful assembly reactions is breathtaking. The conclusions are robust and valid, and well integrated with existing knowledge about regulation of replisome disassembly. The manuscript is suitable for publication in Nature as it addresses multiple outstanding discrepancies in the field. It will catalyse further investigations into this essential process, and influence investigations of how other DNA repair, replication or integration proteins are regulated by DNA association and the ubiquitination machinery.

Minor points for improvement:

Line 73: Reconstructions at 3-4Å resolution – but in the figure the examples are at 4 or 4.3 Å, while the spread in Fig. S1h is 3-7 Å depending on the region of the protein. The findings should be reported more correctly in the main text.

Fig. 1e: Show all of Ctf4 in the figure for clarity.

Fig. 1f: It would be helpful if the orientation and size of the panels e and f were identical.

Fig. 1g: It would be helpful to have a full side-by-side model of the three conformations in a supporting figure, plus a rmsd calculation for the conclusion on lines 139-41 that no conformational changes occur between the three states.

Referee #2:

The manuscript by Jenkyn-Bedford et al. entitled "A conserved mechanism for regulating replisome disassembly in eukaryotes", presents cryo-EM structures of budding yeast and human replisomes in complex with the ubiquitin ligases that are responsible for ubiquitynation of the replisome upon termination of DNA replication.

Research over the last decade revealed that unloading of the replisome from chromatin, upon completion of each replicon duplication, is achieved through a complex and highly regulated

mechanism. In all eukaryotes, upon convergence of the replication forks, one subunit (Mcm7) of the replicative helicase, CMG complex, becomes ubiquitynated, allowing for its unfolding by p97/Cdc48 segregase/unfoldase and its removal from chromatin. In yeast Mcm7 is ubiquitynated by SCF-Dia2, while in higher eukaryotes by Cul2-LRR1 ubiquitin ligase.

The key question in the field was to unravel how these ubiquitin ligases specifically target only the post-termination replisome and not the active replisome. It is a very important point from the genome stability perspective, as uncontrolled disassembly of the active replisome would be catastrophic for cell viability. Recent findings in the field, by the authors of the current manuscript and others, revealed that the manner in which the replisome interacts with DNA is the important determinant for Mcm7 ubiquitynation: when the replisome interacts with forked DNA Mcm7 ubiquitynation is inhibited, but interaction with ssDNA, dsDNA or no DNA is permissive for Mcm7 ubiquitynation.

The key remaining questions are: how does the positioning of the lagging strand of the forked DNA structure prevent the ubiquitin ligase from binding the active replisome and whether the same mechanism is utilised across the eukaryotic kingdom, considering that different ubiquitin ligases evolved to deliver this job.

The present manuscript presents cryo-EM structures of the yeast replisome assembled from purified proteins on dsDNA substrate to resemble post-termination conditions and interacting with SCF-Dia2. The authors could resolve the interaction interfaces of Dia2 and the CMG/replisome complex, finding that Dia2 interacts mainly with Mcm3 and Mcm5 within the CMG and Csm3 replisome component. Although the overall structure of the parts of Mcm3 and Mcm5 that interact with Dia2 are almost identical in the post-termination replisome (on dsDNA) and the active replisome (on forked DNA), this exact part of the structure is where the lagging strand of the replication fork is extruded from the active replisome. Importantly, a number of key interaction points identified in the structure were verified by *in vitro* and *in vivo* functional assays.

The authors also resolved the structure of the human replisome assembled from purified proteins with interacting Cul2-LRR1 ubiquitin ligase. Surprisingly, despite differences between Dia2 and LRR1, the mode of LRR1 interaction with the post-termination replisome is very similar to the yeast counterpart. LRR1 binds to Mcm3/5 but also other parts of the replisome – for example AND-1. The functional importance of these interactions was, however, not verified biochemically or *in vivo*.

In general, this is a very well-written manuscript with masses of exciting and important structural insights that are solving the key questions in the replication termination field. There are a few points that in my mind would improve the manuscript.

Major points:

1. It would be nice to see how the mutations introduced by authors into Mcm3/5 and Dia2 actually affect interaction of SCF-Dia2 with the yeast replisome *in vitro* and *in vivo*. The authors show that Mcm7 ubiquitynation is affected upon introduction of these mutations, but short chains or multi-monoubiquitynation on Mcm7 is still formed (e.g. Fig. 7d,h and Fig. 2c). Does this reflect weaker interaction of SCF-Dia2 with the replisome or a different mode of its action? Do the authors see less mutant SCF-Dia2 interacting with reconstituted replisome *in vitro*? Or less SCF-Dia2 interacting with mutant Mcm3/5 replisome? The authors show IP of Sld5 from G1-arrested cells. If this IP was repeated in S phase, would mutant SCF-Dia2 not interact with the replisome *in vivo*?
2. The authors present assays to functionally verify identified interactions between Mcm3/5 and Dia2, but the interaction platform between Csm3 and Dia2 is not verified; is this functionally important?

3. Previous work from the same group revealed that in budding yeast Ctf4 and Mrc1 components of the replisome are essential for binding of SCF-Dia2 to the replisome. The structure presented here does not identify interaction points between Ctf4 and Mrc1 with SCF-Dia2. On the other hand, Csm3 was shown previously not to be essential for SCF-Dia2 binding to reconstituted replisome. Can authors explain this discrepancy?

4. The authors use here reconstituted human replisome, the structure of which is under revision elsewhere. I did not find in either manuscript evidence whether this reconstituted assembly has the helicase activity expected from CMG and replisome. Does it represent a properly folded complex?

5. The identified interaction points between Cul2-LRR1 and the human replisome have not been validated. I appreciate that this is difficult *in vivo*, but could the authors show that *in vitro* reconstitution of the human replisome, mutation of a few key residues affects Cul2-LRR1's ability to interact with the replisome?

6. The authors nicely show that in the yeast structure the positioning of Mcm3 and Mcm5 does not differ between structures on forked and dsDNA (Fig. 1g). Is the same true for the human structure?

7. The authors discuss the position of the lagging strand extruded from the human replisome structure in Fig. 4a; what about the position of the extruded strand in the yeast structure?

8. It has never been shown that Cul2-LRR1 cannot interact *in vitro* with CMG/replisome bound to forked DNA; it is assumed from *in vivo* observations. Can the authors provide evidence that the human replisome on forked DNA cannot bind Cul2-LRR1?

Minor comments:

1. Page 5, lane 147: The authors need to refer to Ext. Data Fig. 5l,m not k,l.

2. Is the arrangement of the HMG-box of AND-1 in the presented structure a result of Cul2-LRR1 binding or binding of dsDNA vs forked DNA?

3. Ext. Data Fig. 5i: Could Mcm5 ZnF be added to the colour legend at the bottom?

4. Ext. Data Fig. 9: There are two panels labelled "j".

5. Ext. Data Fig. 10e: In the labelling of the LRR repeats there are two number 5 but number 2 is missing.

6. Ext. Data Fig. 11c: It would be nice to have a positive control in the presence of Pif1 in the reaction to show that the system can ubiquitylate Mcm7 on DNA.

Referee #3:

In this work by Jenkyn-Bedford et al., entitled "A conserved mechanism for regulating replisome disassembly in eukaryotes", the authors present two novel cryo-EM structures of the yeast and human replisome at the initial stage of disassembly, supported by additional biochemical data. Combined with a third structure of the human replisome in an active DNA replication mode (Jones et al., currently under review) these structures provide novel and uniquely insightful information on the regulation of replisome disassembly when DNA replication is complete. Their structures show that the protein complexes that ubiquitylate the CMG helicase are prevented access by the displaced strand that runs on the outside of the helicase during DNA synthesis. Remarkably, while

the proteins that perform ubiquitylation are different in yeast (SCF^{>Dia2}: Hrt1-Cdc35-Skp1-Dia2) and human (CUL2^{>LRR1}: LRR1-CUL2-ELOB-ELOC-RBX1), they bind to the same area in the CMG helicase and adopt similar structures, suggestive of convergent evolution. The resolution of the cryo-EM maps is high at 3-4 Å resolution, giving confidence in the quality of the models. The manuscript is clearly written and presents the newly gained insights well.

Combined with a large amount of highly skilled work for the preparation of two large multi-protein complexes of more than 20 subunits each makes this work truly outstanding and will have a strong impact on the field of DNA replication and beyond. I therefore strongly support this work and have only a few minor comments listed below.

1) On p. 5, lines 139-141, they conclude that "...termination does not induce conformational changes in the CMG that are important for the regulation of CMG ubiquitylation...". This immediately raises the question what other factors are needed to regulate the ubiquitylation, yet this is only explained on p. 9. It would therefore be helpful if the authors could add a sentence on p. 5, line 141 along the lines of: "Instead, it appears it is the presence or absence of the displaced strand during DNA synthesis that regulates ubiquitylation, which is discussed further below (see Fig 4)."

2) Twice the authors describe the interaction between the ubiquitylation machinery and the CMG helicase: once for the yeast complex (lines 145-158) and once for the human complex (lines 229-241). Both texts are very detailed and full of information that could be too much for a non-structural biologist. It would therefore be helpful if at the end of these sections a short conclusion would be added to summarize their findings. i.e. something like: "Hence, the interaction between SCF^{>Dia2} is extensive and involves direct contacts between the LRR domain of Dia2 and the MCM subunits 3, 5 and 7 close to the entry point of the dsDNA and is predominantly driven by electrostatic interactions."

3) The structures show how the ubiquitylation machinery is interacting with the replisome on dsDNA and how it is prevented access during initiation or active replication (Fig. 4c). Yet, the authors say nothing about the subsequent steps when the CMG is removed from the DNA. What is known about the mechanism: does it involve opening of the helicase (similar to loading); does it involve disassembly of the complex into smaller parts; or does it require protein degradation by the proteasome? It would be good if they could spend a few words on this in the discussion, even if it is to say that little is known.

Minor comments:

4) p. 6, line 190, "... human replisome (Fig3a,b...": Change to: "... human replisome to nnn Å resolution (Fig3a,b...".

5) Fig 1: It is odd that the structure is presented in panel d, when it is more logical as panel b. In panel b (which should become panel c) the legend should state that Cdc53 is not visible/shown in this view. Also, in panel b, the Dia2 subunit is labelled SCF^{>Dia2} while in panels d-f it is labelled simply Dia2. Finally, it should be noted in the legend that the view in panel b is rotated 180 degrees from panel d (which should be changed to panel c and b, respectively).

Author Rebuttals to Initial Comments:

Referee #1:

The manuscript of Jenkyn-Bedford and Jones et al. represents a tour-de-force characterisation of the mechanism of replisome disassembly in eukaryotes. Several structures of yeast and human

replisome-E3 assemblies are presented that reveal the interactions and associations necessary for replisome disassembly and CMG ubiquitylation. Biochemical reconstitutions revealed loss-of-function mutants in Dia2 that fail to stimulate MCM7 ubiquitylation, leading to accumulation of intact CMG in G1 due to failure of disassembly. I applaud the authors for an outstanding work. The extent of the biochemical reconstitutions and careful assembly reactions is breathtaking. The conclusions are robust and valid, and well integrated with existing knowledge about regulation of replisome disassembly. The manuscript is suitable for publication in Nature as it addresses multiple outstanding discrepancies in the field. It will catalyse further investigations into this essential process, and influence investigations of how other DNA repair, replication or integration proteins are regulated by DNA association and the ubiquitination machinery.

We thank the reviewer for their comments and appreciation of our work.

Minor points for improvement:

Line 73: Reconstructions at 3-4Å resolution - but in the figure the examples are at 4 or 4.3 Å, while the spread in Fig. S1h is 3-7 Å depending on the region of the protein. The findings should be reported more correctly in the main text.

As is common for cryo-EM reconstructions, the resolution varies across different regions of our maps, as is clearly displayed in Extended Data Fig. 1h. The resolution of reconstructions reported in the main text are those calculated at the FSC=0.143 criterion for the various individual multi-body refinement reconstructions used in model building (Extended Data Fig. 1f). This is a widely used metric for reporting on the resolution of cryo-EM maps. As various maps were used, it was inappropriate to provide a single average resolution in the main text.

The 4.0 and 4.3 Å resolutions reported in the Model-to-Map correlations (Extended Data Fig. 1g) relate to the quality of the atomic models built into cryo-EM density, calculated from correlation of experimental cryo-EM density maps to model-derived maps, and therefore do not represent the resolutions of the experimental maps. The main text and the figure legends for Extended Data Fig. 1f,2 have now been amended to clarify this point.

Fig. 1e: Show all of Ctf4 in the figure for clarity.

Fig. 1e has been modified to show all of Ctf4 and Cdc53-Hrt1.

Fig. 1f: It would be helpful if the orientation and size of the panels e and f were identical.

The orientations of panels e and f are now identical. The sizes are as close as possible given space constraints.

Fig. 1g: It would be helpful to have a full side-by-side model of the three conformations in a supporting figure, plus a rmsd calculation for the conclusion on lines 139-41 that no conformational changes occur between the three states.

In Fig. 1g, we have now provided RMSD calculations comparing the Dia2-interacting region of MCM at a replication fork (i.e. prior to termination) with both the termination complexes (dsDNA-bound and off DNA). Critically, the calculated RMSD values (1.39 Å comparing fork-bound to dsDNA-bound CMG, and 0.93 Å comparing fork-bound to off-DNA CMG) support our conclusion that *'termination does not induce conformational changes in CMG that are important for the regulation of CMG ubiquitylation by SCF^{Dia2}, either after fork convergence, or when CMG dissociates from DNA'* (lines 133-135).

Outside of the Dia2-interacting region of MCM, there are differences between the replication fork-associated and terminated replisome structures. For example, in the absence of DNA, some DNA-interacting regions of the replisome, such as Tof1-Csm3 and the MCM C-tier motor domains, are poorly resolved. Furthermore, the replisome structure at a replication fork (Baretic *et al.* Mol. Cell, 2020) lacks SCF^{Dia2} and Pol ε, precluding inclusion of these components in the RMSD analysis. Critically, however, these differences are not relevant for the association of SCF^{Dia2} with the replisome. Thus, we feel that a side-by-side comparison of the full complex would not be overly useful, and may confuse the reader. We note that the structure of a replisome at a replication fork has previously been deposited in the PDB (accession code 6SKL), and the structures we observe in the current study (conformations I and II) will similarly be available from the PDB for comparison.

Referee #2:

The manuscript by Jenkyn-Bedford et al. entitled "A conserved mechanism for regulating replisome disassembly in eukaryotes", presents cryo-EM structures of budding yeast and human replisomes in complex with the ubiquitin ligases that are responsible for ubiquitynation of the replisome upon termination of DNA replication.

Research over the last decade revealed that unloading of the replisome from chromatin, upon completion of each replicon duplication, is achieved through a complex and highly regulated mechanism. In all eukaryotes, upon convergence of the replication forks, one subunit (Mcm7) of the replicative helicase, CMG complex, becomes ubiquitynated, allowing for its unfolding by p97/Cdc48 segregase/unfoldase and its removal from chromatin. In yeast Mcm7 is ubiquitynated by SCF-Dia2, while in higher eukaryotes by Cul2-LRR1 ubiquitin ligase.

The key question in the field was to unravel how these ubiquitin ligases specifically target only the post-termination replisome and not the active replisome. It is a very important point from the genome stability perspective, as uncontrolled disassembly of the active replisome would be

catastrophic for cell viability. Recent findings in the field, by the authors of the current manuscript and others, revealed that the manner in which the replisome interacts with DNA is the important determinant for Mcm7 ubiquitynation: when the replisome interacts with forked DNA Mcm7 ubiquitynation is inhibited, but interaction with ssDNA, dsDNA or no DNA is permissive for Mcm7 ubiquitynation.

The key remaining questions are: how does the positioning of the lagging strand of the forked DNA structure prevent the ubiquitin ligase from binding the active replisome and whether the same mechanism is utilised across the eukaryotic kingdom, considering that different ubiquitin ligases evolved to deliver this job.

We note that these are the primary questions we have focussed on in our current manuscript.

The present manuscript presents cryo-EM structures of the yeast replisome assembled from purified proteins on dsDNA substrate to resemble post-termination conditions and interacting with SCF-Dia2. The authors could resolve the interaction interfaces of Dia2 and the CMG/replisome complex, finding that Dia2 interacts mainly with Mcm3 and Mcm5 within the CMG and Csm3 replisome component. Although the overall structure of the parts of Mcm3 and Mcm5 that interact with Dia2 are almost identical in the post-termination replisome (on dsDNA) and the active replisome (on forked DNA), this exact part of the structure is where the lagging strand of the replication fork is extruded from the active replisome. Importantly, a number of key interaction points identified in the structure were verified by in vitro and in vivo functional assays.

The authors also resolved the structure of the human replisome assembled from purified proteins with interacting Cul2-LRR1 ubiquitin ligase. Surprisingly, despite differences between Dia2 and LRR1, the mode of LRR1 interaction with the post-termination replisome is very similar to the yeast counterpart. LRR1 binds to Mcm3/5 but also other parts of the replisome - for example AND-1. The functional importance of these interactions was, however, not verified biochemically or in vivo.

In general, this is a very well-written manuscript with masses of exciting and important structural insights that are solving the key questions in the replication termination field. There are a few points that in my mind would improve the manuscript.

We thank the reviewer for their appreciation of our work.

Major points:

1. It would be nice to see how the mutations introduced by authors into Mcm3/5 and Dia2 actually affect interaction of SCF-Dia2 with the yeast replisome in vitro and in vivo. The authors show that Mcm7 ubiquitynation is affected upon introduction of these mutations, but short chains or multi-monoubiquitynation on Mcm7 is still formed (e.g. Fig. 7d,h and Fig. 2c). Does this reflect weaker interaction of SCF-Dia2 with the replisome or a different mode of its action? Do the authors see less

mutant SCF-Dia2 interacting with reconstituted replisome *in vitro*? Or less SCF-Dia2 interacting with mutant Mcm3/5 replisome? The authors show IP of Sld5 from G1-arrested cells. If this IP was repeated in S phase, would mutant SCF-Dia2 not interact with the replisome *in vivo*?

The most likely explanation for the very low level of ubiquitylation observed (e.g. in Fig. 2c) is that the extensive Dia2^{LRR}-MCM interaction identified in our structure is not completely abolished in the Dia2 13A (and other) mutant(s). In spite of this, our functional *in vitro* and *in vivo* analyses provide compelling support to the notion that the Dia2^{LRR}-MCM interaction is required for CMG ubiquitylation. This is an important point for the regulatory mechanism we describe, as it indicates that steric occlusion of the Dia2^{LRR}-MCM interface by the excluded DNA strand will be sufficient to block replisome disassembly, until the excluded strand is lost upon termination.

Whilst we agree with the reviewer that it would be interesting to examine SCF^{Dia2} recruitment to replication forks *in vivo*, the suggested S-phase IP experiment is unfortunately inherently flawed. We necessarily release CMG from DNA (by benzonase treatment in cell extracts) as part of our immunoprecipitation protocol. In doing this, we remove the excluded DNA strand-based negative regulation that we describe in our current manuscript, which allows for *in vitro* CMG-Mcm7 ubiquitylation in cell extracts (as described in Maric *et al.*, Science, 2014). Thus, any SCF^{Dia2} recruitment observed in these experiments is primarily an *in vitro* phenomenon, and does not address the question of how and when SCF^{Dia2} is recruited *in vivo*.

We have done the *in vitro* experiment suggested by the reviewer (see Extended Data Fig. 11d,e), and find that Mrc1-Ctf4 are able to recruit SCF^{Dia2-13A} to reconstituted CMG-replisome complexes. CMG-Mcm7 ubiquitylation is profoundly defective under the same experimental conditions, indicating the recruitment of SCF^{Dia2} to the replisome is not sufficient *per se* for ubiquitylation. Accordingly (and as we point out in lines 268-275), even if SCF^{Dia2} were recruited to replication forks via Mrc1-Ctf4 during elongation, CMG ubiquitylation (and hence replisome disassembly) would still be prevented, as long as the Dia2^{LRR}-MCM interface were blocked by the excluded strand. Thus, although interesting, our data indicate that the timing of SCF^{Dia2} recruitment is fairly inconsequential for the regulatory mechanism we describe. Furthermore, proper examination of this question would most likely involve single molecule analyses of SCF^{Dia2} recruitment during DNA replication *in vitro*, which is beyond the scope of our current manuscript.

2. The authors present assays to functionally verify identified interactions between Mcm3/5 and Dia2, but the interaction platform between Csm3 and Dia2 is not verified; is this functionally important?

We identified several classes of particles in our cryo-EM datasets that clearly lacked Tof1-Csm3 but retained strong electron density for Dia2 (e.g. Figure 1f, where the absence of DNA destabilised

Tof1-Csm3 binding to CMG), indicating that the small Csm3-Dia2 interface described in our structure is not essential for SCF^{Dia2} association with the replisome. Consistent with this, Tof1-Csm3 is not required for CMG-Mcm7 ubiquitylation in reconstituted reactions using budding yeast proteins (see Deegan *et al.*, eLife, 2020).

3. Previous work from the same group revealed that in budding yeast Ctf4 and Mrc1 components of the replisome are essential for binding of SCF-Dia2 to the replisome. The structure presented here does not identify interaction points between Ctf4 and Mrc1 with SCF-Dia2. On the other hand, Csm3 was shown previously not to be essential for SCF-Dia2 binding to reconstituted replisome. Can authors explain this discrepancy?

In agreement with a large body of previously published work from the Labib lab, Mrc1 and Ctf4 are required for the association of SCF^{Dia2} with CMG under the conditions used for complex assembly in our current study (e.g. see Extended Data Fig. 11d,e). However, as detailed in previous publications (e.g. Baretic *et al.*, Mol Cell, 2020), the majority of Mrc1 and the N-terminal portion of Ctf4 are difficult to visualise using existing cryo-EM methodologies, likely due to the presence of large unstructured and/or flexible regions in these proteins. The 'invisibility' of these proteins explains why we are currently unable to visualise the interaction between the TPR domain of Dia2 and Mrc1-Ctf4 in our structure.

4. The authors use here reconstituted human replisome, the structure of which is under revision elsewhere. I did not find in either manuscript evidence whether this reconstituted assembly has the helicase activity expected from CMG and replisome. Does it represent a properly folded complex?

This response has been redacted as it relates to unpublished material.

5. The identified interaction points between Cul2-LRR1 and the human replisome have not been validated. I appreciate that this is difficult in vivo, but could the authors show that in vitro reconstitution of the human replisome, mutation of a few key residues affects Cul2-LRR1's ability to interact with the replisome?

The resolution of our cryo-EM structure at the interfaces between LRR1 and the replisome is high, typically around 3 Å, which enabled unambiguous modelling of the specific amino acid contacts important for these interactions. Consequently, because we have been able to model 92 % of LRR1, our structure alone strongly indicates that the two sites we identify - one involving the LRR1 PH domain and the other the C-terminal end of the LRR1 LRR domain - are the main attachment points for Cul2^{LRR1} in the replisome. The interaction of the LRR1 PH domain with TIMELESS is consistent with a recent study showing that TIMELESS is involved in CMG ubiquitylation in *C. elegans* (Xia *et al.*, EMBO J., 2021). Furthermore, although the same study showed that TIM-1_TIPN-1 stimulates Mcm7 ubiquitylation, ubiquitylation was still observed in reactions comprising only CMG, Cul-2^{LRR-1} and the ubiquitylation machinery. Importantly, ubiquitylation did not occur when Cul-2^{LRR-1} was replaced by

Cul-2^{VHL-1}, demonstrating that it was dependent on LRR-1, very likely via the extensive interaction interface we have discovered between LRR1^{LRR} and MCM.

Notably, the C-terminal end of the cullin arm and its associated RING-box protein are located in a very similar position in our human and yeast structures, opposite the previously reported ubiquitylation sites on the N-tier of MCM7. Crucially, this positioning, which is supported by ubiquitylation site mapping data from various model systems (Maric *et al.*, Cell Rep., 2017, Low *et al.*, Genes Dev., 2020, Deegan *et al.*, eLife, 2020), is a direct consequence of the manner in which the Dia2^{LRR} and LRR1^{LRR} domains interact with MCM, and thereby provides further strong evidence in support of the LRR1^{LRR}:MCM interaction we have discovered.

This text has been redacted as it relates to unpublished material.

6. The authors nicely show that in the yeast structure the positioning of Mcm3 and Mcm5 does not differ between structures on forked and dsDNA (Fig. 1g). Is the same true for the human structure?

The human replisome-E3 ligase complex described in the current manuscript was assembled in the presence of the non-hydrolysable ATP analogue AMP-PNP on a DNA structure lacking a 5' flap. Because of this, human CMG did not translocate onto the dsDNA region of the substrate, and therefore the C-tier motor domains were bound to single-stranded DNA. This is different to our yeast structure, where the presence of hydrolysable ATP facilitated CMG translocation onto dsDNA, which in turn allows the comparisons depicted in Fig. 1g. There are several technical reasons why we used this approach to prepare our human replisome complex. Our initial cryo-EM datasets of human replisomes with CUL2^{LRR1} indicated that the occupancy of LRR1 was lower than Dia2 in the yeast replisome. Given that only a fraction of yeast replisomes translocated onto dsDNA in the presence of ATP, we thought it would be challenging to obtain a sufficient number of particles that both contained Cul2^{LRR1} and had translocated onto dsDNA. Therefore, to maximise the chance of obtaining a sufficient number of LRR1-containing particles for high resolution reconstructions, we prepared the human replisome sample in the presence of AMP-PNP.

Despite these differences in sample preparation, the key point is that the human replisome structure still lacked the excluded strand of the replication fork that is required to suppress Mcm7 ubiquitylation. Indeed, work from the Walter lab showed that translocation of *Xenopus* CMG onto dsDNA is not required for replisome disassembly, because, when DNA synthesis was inhibited with aphidicolin, replisomes bound around ssDNA were efficiently disassembled after fork convergence (Low *et al.*, *Genes Dev.*, 2020). Furthermore, given the remarkable similarity in the way yeast and human CMG complexes engage replication fork DNA, both in the C-tier and at the fork junction (Baretic *et al.*, *Mol Cell*, 2020, Yuan *et al.*, *Nat Comm.*, 2020, Rzechorzek *et al.*, *NAR*, 2020, Jones *et al.*, in revision), we consider it very unlikely that human CMG will display significant conformational changes upon dsDNA engagement that are not observed in yeast.

7. The authors discuss the position of the lagging strand extruded from the human replisome structure in Fig. 4a; what about the position of the extruded strand in the yeast structure?

Prior to this manuscript and Jones *et al.* (in revision), cryo-EM density attributed to the excluded strand had only ever been recovered in structures of translocating *Drosophila* CMG, via data processing using the CryoSparrc program (Eickhoff *et al.*, *Cell Reports*, 2019). In our other manuscript (Jones *et al.*, in revision), similar data processing allowed us to observe cryo-EM density in the same position (i.e. between the MCM3 and MCM5 ZnFs) for the human replisome. We unambiguously attribute this density to the excluded strand in Fig. 4a of our current manuscript.

We have been unable to resolve the excluded strand upon re-processing a prior yeast replisome dataset (Baretic *et al.*, *Mol. Cell*, 2020) using the CryoSparrc approach. However, in yeast structures where a single nucleotide of the unpaired lagging strand template (i.e. the excluded strand) has been resolved, this strand points directly towards the channel between the Mcm3 and Mcm5 ZnFs (Yuan *et al.*, *Nat. Commun.*, 2020, Baretic *et al.* *Mol. Cell*, 2020). Notably, this channel is lined by several highly conserved basic residues proposed to interact directly with the excluded strand (Yuan *et al.*, *Nat. Commun.*, 2020). Furthermore, the position of the incoming parental dsDNA and the point of strand separation are identical between yeast and metazoan CMG, supporting the idea that the path of the excluded strand is conserved between yeast and metazoa.

8. It has never been shown that Cul2-LRR1 cannot interact in vitro with CMG/replisome bound to forked DNA; it is assumed from in vivo observations. Can the authors provide evidence that the human replisome on forked DNA cannot bind Cul2-LRR1?

In reality, experiments with CMG and short linear oligonucleotide-based forked DNA substrates are inherently limited, as CMG is in equilibrium with the DNA under such conditions, and free CMG is an excellent substrate for CUL2^{LRR1} binding and ubiquitylation, when not bound at a replication fork

(e.g. see Low *et al.*, Genes Dev., 2020, Xia *et al.*, EMBO J, 2021). Importantly however, work in frog egg extracts from the Gambus and Walter laboratories has indicated that the recruitment of CUL2^{LRR1} to replication forks is blocked during elongation (Sonneville *et al.*, Nat. Cell Biol., 2017, Dewar *et al.*, Genes Dev., 2017). CUL2^{LRR1} recruitment (and CMG-Mcm7 ubiquitylation) is only permitted upon loss of the excluded strand, either when CMG is not bound to DNA (Low *et al.*, Genes Dev., 2020, Xia *et al.*, EMBO J, 2021), or when CMG is bound around dsDNA, either after fork convergence (Dewar *et al.*, Genes Dev., 2017) or when CMG has translocated over a nick in the excluded strand (Vrtis *et al.*, Mol Cell, 2021). Consistent with and building on this extensive body of data, our new work provides compelling evidence that the regulation of CUL2^{LRR1} by the excluded strand is direct, and works via steric inhibition of the LRR-MCM interface identified in our structures.

Minor comments:

1. Page 5, lane 147: The authors need to refer to Ext. Data Fig. 5l,m not k,l.

We have changed this.

2. Is the arrangement of the HMG-box of AND-1 in the presented structure a result of Cul2-LRR1 binding or binding of dsDNA vs forked DNA?

The HMG-box of AND-1 is positioned alongside ELOC and LRR1 and appears to make little, if any, contact with CMG subunits. Therefore, because the AND-1 HMG box is connected to the Sep-B domain by a long flexible linker, the arrangement of the AND-1 HMG-box observed in the presence of Cul2^{LRR1} is almost certainly due to protein:protein interactions with subunits of Cul2^{LRR1}.

3. Ext. Data Fig. 5i: Could Mcm5 ZnF be added to the colour legend at the bottom?

Mcm5 ZnF has been added to the colour legend for Ext. Data Fig. 5l.

4. Ext. Data Fig. 9: There are two panels labelled “j”.

We have changed this.

5. Ext. Data Fig. 10e: In the labelling of the LRR repeats there are two number 5 but number 2 is missing.

We have changed this.

6. Ext. Data Fig. 11c: It would be nice to have a positive control in the presence of Pif1 in the reaction to show that the system can ubiquitylate Mcm7 on DNA.

The omission of multiple critical replisome proteins in Ext. Data Fig. 11c severely inhibits replication fork elongation, which means that forks will not converge and terminate, even in the presence of Pif1. Thus, the best positive control for ubiquitylation in this experiment is replisomes that have been released from DNA by benzonase treatment.

Referee #3:

In this work by Jenkyn-Bedford et al., entitled "A conserved mechanism for regulating replisome disassembly in eukaryotes", the authors present two novel cryo-EM structures of the yeast and human replisome at the initial stage of disassembly, supported by additional biochemical data. Combined with a third structure of the human replisome in an active DNA replication mode (Jones et al., currently under review) these structures provide novel and uniquely insightful information on the regulation of replisome disassembly when DNA replication is complete. Their structures show that the protein complexes that ubiquitylate the CMG helicase are prevented access by the displaced strand that runs on the outside of the helicase during DNA synthesis. Remarkably, while the proteins that perform ubiquitylation are different in yeast (SCF^{Dia2} : Hrt1-Cdc35-Skp1-Dia2) and human ($CUL2^{LRR1}$: LRR1-CUL2-ELOB-ELOC-RBX1), they bind to the same area in the CMG helicase and adopt similar structures, suggestive of convergent evolution. The resolution of the cryo-EM maps is high at 3-4 Å resolution, giving confidence in the quality of the models. The manuscript is clearly written and presents the newly gained insights well.

Combined with a large amount of highly skilled work for the preparation of two large multi-protein complexes of more than 20 subunits each makes this work truly outstanding and will have a strong impact on the field of DNA replication and beyond. I therefore strongly support this work and have only a few minor comments listed below.

We thank the reviewer for their comments and for their appreciation of our work.

1) On p. 5, lines 139-141, they conclude that "...termination does not induce conformational changes in the CMG that are important for the regulation of CMG ubiquitylation...". This immediately raises the question what other factors are needed to regulate the ubiquitylation, yet this is only explained on p. 9. It would therefore be helpful if the authors could add a sentence on p. 5, line 141 along the lines of: "Instead, it appears it is the presence or absence of the displaced strand during DNA synthesis that regulates ubiquitylation, which is discussed further below (see Fig 4)."

We appreciate the reviewer's point, but respectfully take a different view. The key conclusion that CMG ubiquitylation is regulated via steric occlusion of the LRR-MCM interface by the excluded DNA strand is dependent on a range of functional and structural insights presented after line 141 (now line 133). Of particular importance are the functional analyses described in Fig. 2 and Extended Data Fig. 7, the solving of the human replisome-E3 ligase structure in Fig. 3, and the identification of the path of the excluded strand in Fig. 4a. For these reasons, we feel that it is appropriate for the mechanism of regulation to only be revealed at the end of the manuscript (in Fig. 4b-c), once appropriate supporting data for our model has been described in the preceding figures.

2) Twice the authors describe the interaction between the ubiquitylation machinery and the CMG helicase: once for the yeast complex (lines 145-158) and once for the human complex (lines 229-241). Both texts are very detailed and full of information that could be too much for a non-structural biologist. It would therefore be helpful if at the end of these sections a short conclusion would be added to summarize their findings. i.e. something like: "Hence, the interaction between SCF^{Dia2} is extensive and involves direct contacts between the LRR domain of Dia2 and the MCM subunits 3, 5 and 7 close to the entry point of the dsDNA and is predominantly driven by electrostatic interactions."

We thank the reviewer for this suggestion and have modified the text describing the MCM-LRR interfaces for yeast (lines 139-145) and human (lines 193-223) accordingly.

3) The structures show how the ubiquitylation machinery is interacting with the replisome on dsDNA and how it is prevented access during initiation or active replication (Fig. 4c). Yet, the authors say nothing about the subsequent steps when the CMG is removed from the DNA. What is known about the mechanism: does it involve opening of the helicase (similar to loading); does it involve disassembly of the complex into smaller parts; or does it require protein degradation by the proteasome? It would be good if they could spend a few words on this in the discussion, even if it is to say that little is known.

After ubiquitylation, the CMG-replisome is disassembled through the combined action of the Cdc48/p97 'segregase' complex and its associated co-factors Ufd1-Npl4. As the reviewer points out, numerous possibilities could be envisaged for how the disassembly reaction might work. Importantly, however, biochemical reconstitutions of Cdc48-dependent disassembly of ubiquitylated CMG (using proteins from various organisms) have now shed significant light on this issue (see Deegan *et al.*, *eLife*, 2020 and Xia *et al.*, *EMBO J*, 2021). The significant mechanistic advance arising from these studies was to show that Cdc48 disassembles ubiquitylated CMG by unfolding (only) the ubiquitylated Mcm7 subunit, via translocation of ubiquitylated Mcm7 through the central channel of the Cdc48 hexamer. We have now added a sentence describing this mechanism to our introduction (lines 44-45)

Minor comments:

4) p. 6, line 190, "... human replisome (Fig3a,b...": Change to:
"... human replisome to nnn Å resolution (Fig3a,b...".

We have amended the text here to state: "...determine a high resolution structure of CUL2^{LRR1} in the human replisome (Fig.3a,b...". Due to the wide range of different resolutions reported for the various reconstructions used to build the atomic model, we do not feel it is appropriate to provide a single average representative value. Indeed, the best estimate for an "average" resolution for the full complex is 2.8 Å, which does not accurately reflect the resolution at the novel interfaces between CUL2^{LRR1} and the core replisome and may mislead readers. A complete description of the resolution range of each individual sub-refinement would be lengthy. We cite Extended Data Fig. 8, 9a-e, which provide details on resolution range and local resolution.

5) Fig 1: It is odd that the structure is presented in panel d, when it is more logical as panel b. In panel b (which should become panel c) the legend should state that Cdc53 is not visible/shown in this view. Also, in panel b, the Dia2 subunit is labelled SCF^{Dia2} while in panels d-f it is labelled simply Dia2. Finally, it should be noted in the legend that the view in panel b is rotated 180 degrees from panel d (which should be changed to panel c and b, respectively).

In our description of a terminated replisome, it was important to first identify the subset of replisome complexes which had successfully translocated onto double-stranded DNA (panels b and c), prior to describing in detail the architecture of a terminated replisome (panel d). In panel b, the different subunits of SCF^{Dia2} were not distinguished as further processing (described in the main text) was required to improve the resolution of this region, in order to distinguish the exact positions of the various subunits (Dia2, Skp1 and Cdc53). This is why the E3 ligase is identified as SCF^{Dia2} in panel b, whilst individual subunits were identified in later panels. For these reasons, we feel it would be most appropriate to retain the original ordering of panels in Fig. 1.

We note that the relationship between panels b and d is not a simple 180 degree rotation, and we believe an in depth description of how these panels relate to one another would not be particularly beneficial.

[Redacted]

This text has been redacted as it relates to unpublished material.

This text has been redacted as it relates to unpublished material.

Reviewer Reports on the First Revision:

Referee #2:

Thank you for response to my comments and inclusion of some more explanations in the text of the manuscript.

I understand the limitations of what can be resolved etc. and take the arguments put forward. Thank you.

My one remaining comment:

In case of my point 6, I asked for comparison of the human Mcm3/5 platform (that interacts with LRR1) between structure on forked and dsDNA. The authors explain that they cannot provide this as their terminated structure is not on dsDNA but on a fork without a flap. I see the point. Can we see the overlay of this interaction platform between structure on forked DNA and flap-less fork DNA (as the best approximation to dsDNA)? It would suggest whether protruding lagging strand leads to any changes in the structure of this region.

Referee #3:

The authors have answered all queries satisfactorily. I have no further comments

Author Rebuttals to First Revision:

Referee #2:

Thank you for response to my comments and inclusion of some more explanations in the text of the manuscript.

I understand the limitations of what can be resolved etc. and take the arguments put forward. Thank you.

My one remaining comment:

In case of my point 6, I asked for comparison of the human Mcm3/5 platform (that interacts with LRR1) between structure on forked and dsDNA. The authors explain that they cannot provide this as their terminated structure is not on dsDNA but on a fork without a flap. I see the point. Can we see the overlay of this interaction platform between structure on forked DNA and flap-less fork DNA (as the best approximation to dsDNA)? It would suggest whether protruding lagging strand leads to any changes in the structure of this region.

We have included the requested comparison as Extended Data Fig. 10o. The figure shows that there are no significant differences in the regions of MCM3 and MCM5 that interact with LRR, when a 5'-flap is present or not.

Referee #3:

The authors have answered all queries satisfactorily. I have no further comments